# DECOMPOSING PROMPTS: DISCOVERING REUSABLE SCAFFOLDS AND TASK-SPECIFIC RESIDUALS

## ABSTRACT

Traditional automatic prompt optimization methods treat prompts as monolithic text blocks, necessitating costly, from-scratch optimization for each new task and precluding knowledge reuse across related tasks. We revisit this monolithic paradigm by proposing an approach where prompts are decomposed into two components: a reusable, domain-invariant **Instructional Scaffold** that captures high-level task structure, and a concise **Task-specific Residual** for fine-grained adaptation. We introduce **DiSR** (**D**iscovering **i**nstructional **S**caffolds and **R**esiduals), a two-stage algorithm inspired by the Minimum Description Length (MDL) principle, which motivates decomposing prompts into reusable components. This approach systematically discovers the components, allowing for efficient knowledge reuse and task-specific adaptation. In contrast to existing methods like OPRO and AutoPrompt, DiSR enables task-level knowledge reuse, improving performance and reliability across different tasks. Extensive experiments demonstrate that DiSR not only achieves competitive accuracy but also generates better-calibrated confidence estimates, especially on large models and in professional domains. This compositional hypothesis is validated through semantic analysis, revealing that optimized prompts form distinct, scaffold-centered clusters in their embedding space. Our findings establish a compositional view of prompt engineering, facilitating more robust, interpretable, and reusable optimization strategies.

## 1 INTRODUCTION

Large Language Models (LLMs) have demonstrated remarkable capabilities across a wide range of tasks, unlocked by effective prompts (Brown et al., 2020). Automated prompt optimization (APO) methods, which use the LLM as an optimizer to discover high-performance prompts, have emerged to replace manual tuning. Seminal works, including LLM-as-optimizer methods like OPRO (Yang et al., 2024) and contemporaneous approaches (Zhou et al., 2022; Pryzant et al., 2023; Wang et al., 2023), as well as gradient-based methods like AutoPrompt (Shin et al., 2020), show that machine-generated instructions can outperform human-crafted ones on standard benchmarks (Chang et al., 2024). However, these methods suffer from a critical flaw: they treat prompts as monolithic, opaque units, meaning that each prompt is treated as a single, indivisible block of text, requiring expensive optimization to restart *de novo* for each new task. This prevents the reuse of effective patterns and incurs substantial costs—a challenge conceptually akin to meta-learning (Finn et al., 2017).

To address this limitation, we propose leveraging the internal compositional structure of prompts. Distinct from decomposing tasks into sequential workflows (Khot et al., 2022), we target the prompt's *internal semantic structure* to enable knowledge reuse. While instruction tuning (Wei et al., 2021) suggests reusable patterns, current APO methods lack mechanisms to discover them. Inspired by the Minimum Description Length (MDL) principle (Rissanen, 1978), we hypothesize that optimal prompts balance a shared structure with task-specific refinements. This compact representation preserves domain knowledge while facilitating fine-grained adaptation (analogous to parameter-efficient tuning (Hu et al., 2021)), leading to our central hypothesis: an optimal prompt decomposes into a domain-invariant structural component and a task-specific refinement.

In this paper, we formalize this concept by decomposing prompts into two components: an **Instructional Scaffold** and a **Task-specific Residual**. The scaffold serves as a reusable semantic and structural backbone for a domain (e.g., "clinical diagnosis"), enabling communication with the

LLM. The residual is a concise adaptation that fine-tunes the scaffold for a specific task (e.g., "given a patient history, provide the most likely diagnosis" vs. "given a panel of lab results, explain the most likely underlying condition"). To operationalize and validate this theory, we introduce **DiSR** (**D**iscovering **i**nstructional **S**caffolds and **R**esiduals), a two-stage framework. The first stage, *Scaffold Discovery*, identifies the shared scaffold by optimizing across diverse source tasks. The second stage, *Residual Learning*, rapidly learns the minimal, task-specific residual from the scaffold.

Our contributions are fourfold:

- We introduce a new conceptual framework for prompt engineering, framing optimization as a decomposition problem guided by the MDL principle.
- We propose *Instructional Scaffolds* and *Task-specific Residuals* as fundamental building blocks of prompts, shifting the focus from monolithic strings to modular components.
- We present **DiSR**, a two-stage algorithm designed to discover and leverage components.
- We provide extensive empirical validation, demonstrating competitive performance and better-calibrated confidence estimates over SOTA baselines, and, via semantic visualization, showing that optimized prompts exhibit a scaffold-centered modular structure.

## 2 RELATED WORK

Our work builds upon Automated Prompt Optimization (APO), which aims to automate manual prompt crafting. This field is characterized by two major approaches: leveraging LLMs as optimizers, as seen in OPRO and Automatic Prompt Engineer (Yang et al., 2024; Zhou et al., 2022), and gradient-based token search methods like AutoPrompt (Shin et al., 2020). While effective, these methods treat prompts as monolithic, unstructured text, preventing knowledge reuse across tasks. We depart from this paradigm by introducing a structured, compositional approach. Conceptually, our framework translates the principles of Parameter-Efficient Fine-Tuning (PEFT) (Houlsby et al., 2019; Hu et al., 2021) into the prompt engineering domain. Similar to how PEFT adapts a frozen model by tuning a small set of adapter parameters, **DiSR** learns a concise **Task-specific Residual** to adapt a robust, shared **Instructional Scaffold**, bringing similar efficiency and modularity to prompt optimization. Our focus on decomposing a human-interpretable prompt distinguishes our method from continuous soft prompt optimization, as in Prompt Tuning (Lester et al., 2021), and modular prompting frameworks such as Decomposed Prompting (Khot et al., 2022), which decompose complex tasks into sub-task prompts at the workflow level rather than the internal structure of a single prompt, as well as manual curation efforts like the PromptSource library (Bach et al., 2022).

The theoretical underpinnings of our compositional approach are rooted in information theory and cognitive science. We adopt an MDL-guided view of the Minimum Description Length (MDL) principle (Rissanen, 1978), which motivates compressing related prompts by encoding a shared structure (the scaffold) and minimal, task-specific deviations (the residuals). This perspective is complemented by Schema Theory from cognitive science, which posits that humans organize and reuse knowledge in structured, abstract frameworks analogous to our scaffolds (Bartlett, 1995; Rumelhart, 2017). Recent work has demonstrated LLMs' emergent human-like analogical reasoning capabilities (Webb et al., 2023). While prior work has focused on the engineering aspects of APO, within this line of work our approach synthesizes these complementary principles from formal theory and cognitive science into an algorithm for structured prompt discovery and reuse.

## 3 METHODOLOGY

### 3.1 ON THE DEFINITION OF A TASK DOMAIN

Our framework's central hypothesis posits a shared instructional structure within a task *domain*. We operationally define a domain as a set of tasks that share a common goal, format, evaluation metric, and reasoning process (e.g., multiple-choice question answering in fields such as commonsense reasoning, medicine, law, or finance). Although this definition is heuristic, our experiments confirm that it successfully yields reusable scaffolds across the evaluated domains. Concretely, we enforce a fixed input/output format and metric, while constructing $\mathcal{S}$ and $\mathcal{T}$ from sub-tasks that vary in topic

Figure 1: Overview of the DiSR framework, which decomposes prompt optimization into two stages. **Stage 1 (Scaffold Discovery):** A shared, domain-invariant Instructional Scaffold ($P_{\text{scaffold}}$) is learned by optimizing across multiple source tasks. **Stage 2 (Residual Learning):** The scaffold serves as a high-quality starting point, and the optimization loop performs task-specific **adaptation**, represented by the Task-specific Residual ($\Delta P_k$), yielding the final specialized prompt ($P_k^*$).

or reasoning type; Section 4 and the appendix further analyze how scaffold quality depends on the number and diversity of these source tasks.

## 3.2 PRELIMINARIES AND PROBLEM FORMULATION

Let a domain be defined by a set of source tasks $\mathcal{S} = \{\mathcal{S}_1, \dots, \mathcal{S}_\kappa\}$ and target tasks $\mathcal{T} = \{\mathcal{T}_1, \dots, \mathcal{T}_\tau\}$. Each target task $\mathcal{T}_k$ is associated with a dataset $D_k = \{(q_n, a_n)\}_{n=1}^{N_k}$, where $q_n$ represents an input query (e.g., a question or prompt), and $a_n$ represents the corresponding desired output (e.g., the correct answer or model response), which serves as the reference for evaluating prompt quality and guiding the optimization process.

Traditional prompt optimization aims to find a task-specific prompt $P_k^*$ for each target task $\mathcal{T}_k$ that maximizes a performance metric $\phi$:

$$P_k^* = \arg\max_P \phi(P, D_{\mathcal{T}_k}) \tag{1}$$

This formulation (Eq. 1) treats each prompt as an independent entity, leading to redundant optimization efforts across related tasks.

Our work reformulates this problem by assuming that optimal prompts for related tasks share reusable structure, enabling the reuse of instructional scaffolds across tasks. Instead of optimizing $\tau$ independent prompts, we aim to discover a domain-level **Instructional Scaffold** ($P_{\text{scaffold}}$) and, for each task, a minimal **Task-specific Residual** $\Delta P_k$ such that $P_k^* = P_{\text{scaffold}} \oplus \Delta P_k$.

This perspective can be formalized in an MDL-inspired objective that trades off model complexity against empirical error:

$$L_{\text{MDL}} = L(P_{\text{scaffold}}) + \sum_{k=1}^{\tau} L(\Delta P_k) + \sum_{k=1}^{\tau} L_{\text{err}}(P_k^*, D_{\mathcal{T}_k}), \tag{2}$$

where $L(\cdot)$ denotes a description-length proxy (e.g., the token length of the instruction) and $L_{\text{err}}$ measures task-specific error (e.g., misclassification rate). Our two-stage procedure can be viewed as a practical, greedy approximation to minimizing $L_{\text{MDL}}$: Stage 1 learns a single scaffold that amortizes $L(P_{\text{scaffold}})$ across source tasks, while Stage 2 adds minimal residuals $\Delta P_k$ to reduce $L_{\text{err}}(P_k^*, D_{\mathcal{T}_k})$ on individual target tasks.

## 3.3 THE DiSR FRAMEWORK: A DECOMPOSITIONAL APPROACH

To address this reformulated problem, we propose **DiSR**, a two-stage framework that materializes the hypothesized decomposition. As illustrated in Figure 1, DiSR separates the discovery of the domain-invariant scaffold from the learning of the task-specific residual.

Conceptually, we model the final prompt $P_k^*$ as the result of a targeted adaptation process starting from the scaffold $P_{\text{scaffold}}$. This procedural view is central to our hypothesis: rather than optimizing from scratch, DiSR leverages a shared scaffold that captures domain-specific knowledge and adapts it for each new task with minimal additional information. The transformation from scaffold to task-specific prompt is represented as:

$$P_{\text{scaffold}} \xrightarrow{\text{Adapt}(D_{\mathcal{T}_k})} P_k^* \tag{3}$$

Here, the Adapt operator represents the Stage 2 optimization process, using data $D_{\mathcal{T}_k}$ to refine the scaffold into the final prompt $P_k^*$. The **Task-specific Residual** ($\Delta P_k$) is the semantic change embodied by this transformation, and we write the decomposition explicitly as

$$P_k^* = P_{\text{scaffold}} \oplus \Delta P_k, \tag{4}$$

where $\oplus$ denotes composing the scaffold with a small set of task-specific edits (e.g., additions or localized modifications to the scaffold template).

Operationally, the framework uses $P_{\text{scaffold}}$ as a high-quality starting point for the optimization in Stage 2. The resulting prompt, $P_k^*$, incorporates the scaffold's foundational knowledge plus the incremental refinements from the residual transformation.

### 3.4 STAGE 1: SCAFFOLD DISCOVERY

The objective is to distill a domain-invariant Instructional Scaffold by optimizing a single prompt to achieve strong average performance across a diverse set of source tasks. This encourages the scaffold to capture essential domain knowledge while remaining robust and adaptable to new, related tasks.

**Input:** A set of source datasets $\{D_{\mathcal{S}_1}, \ldots, D_{\mathcal{S}_\kappa}\}$ and an initial, generic prompt $P_0$.

**Process:** We use an iterative optimization loop (Sec. 3.6) where each candidate prompt $P$ is evaluated against all source datasets. Its score $s(P)$ is the average performance:

$$s(P) = \frac{1}{\kappa} \sum_{i=1}^{\kappa} \phi(P, D_{\mathcal{S}_i}) \tag{5}$$

This cross-task objective forces the optimizer to retain only shared, domain-essential semantics.

**Output:** The highest-scoring prompt is designated as the Instructional Scaffold:

$$P_{\text{scaffold}} = \arg\max_P s(P) \tag{6}$$

### 3.5 STAGE 2: RESIDUAL LEARNING

Given the scaffold, Stage 2 learns the minimal adaptation required for a target task, corresponding to the Task-specific Residual transformation. This adaptation process tailors the scaffold to the specific requirements of the task while preserving underlying domain knowledge.

**Input:** The discovered scaffold $P_{\text{scaffold}}$ and a target task dataset $D_{\mathcal{T}_k}$.

**Process:** The optimization loop is re-initiated with two distinctions: the search starts from $P_{\text{scaffold}}$, and evaluation is *only* on the target dataset $D_{\mathcal{T}_k}$. The objective is thus:

$$P_k^* = \arg\max_{P'} \phi(P', D_{\mathcal{T}_k}) \quad \text{(initialized from } P_{\text{scaffold}}) \tag{7}$$

In Stage 2, the meta-prompt (Figure 2) conditions on $P_{\text{scaffold}}$ and asks the Proposer LLM to preserve it while proposing small, task-specific edits, so that the learned residual $\Delta P_k$ remains concise.

**Output:** The final, task-specialized prompt $P_k^*$.

### 3.6 CORE OPTIMIZATION PROCESS

Both DiSR stages are powered by the same optimization engine, adapted from the "LLM as optimizer" paradigm in OPRO (Yang et al., 2024). This choice isolates the effects of our framework's

novel problem formulation, attributing performance gains to the decompositional approach itself, rather than a new optimization algorithm. The engine has three components:

**Proposer LLM:** A generator model that proposes new candidate instructions at each iteration based on a meta-prompt.

**Scorer LLM:** A model acting as the objective function $\phi$, which executes prompts on input queries and evaluates their accuracy against the ground truth.

**Meta-Prompt:** A carefully crafted prompt that instructs the Proposer. As shown in Figure 2, it includes the task objective, format examples, and a history of prior prompts and their scores to guide the search for better instructions; in Stage 2, the same template provides $P_{\text{scaffold}}$ and emphasizes making minimal, task-specific edits relative to it rather than rewriting the prompt from scratch.

For a fair comparison, all APO baselines (including OPRO) reuse the same Proposer/Scorer pair, meta-prompt template, and initial manual prompt $P_0$; the only difference is that they optimize each task-specific prompt directly from $P_0$ instead of from $P_{\text{scaffold}}$.

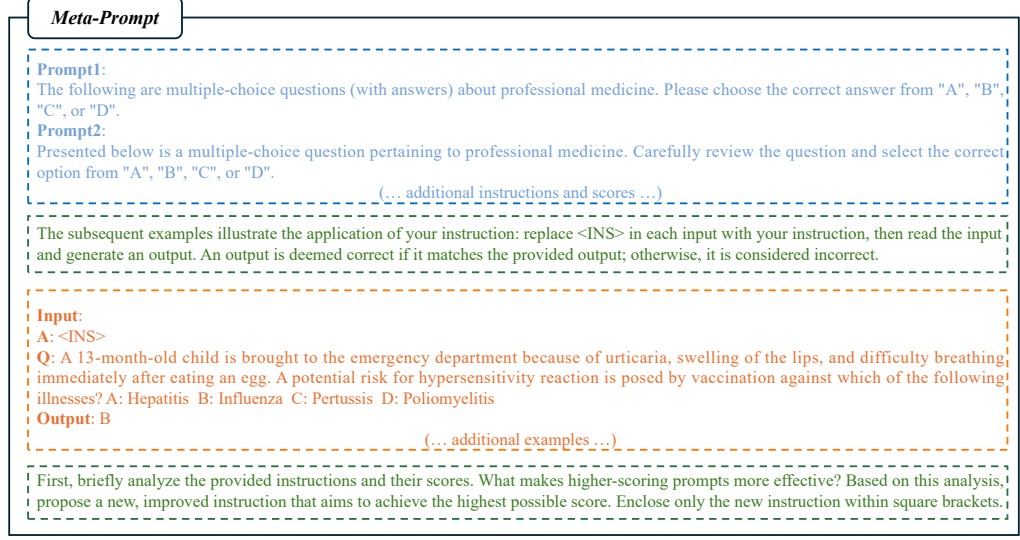

Figure 2: Structure of the Meta-Prompt guiding the Proposer LLM: (1) **Optimization History**, a list of past prompts and their scores; (2) **Task Schema**, an example input/output format; and (3) **Reflective Instruction**, directing the LLM to analyze past data before proposing new prompts; in Stage 2, the meta-prompt also conditions on the discovered scaffold and asks for small, task-specific refinements rather than wholesale replacements.

## 4 EXPERIMENTS

### 4.1 EXPERIMENTAL SETUP

**Datasets** To ensure a robust evaluation, we utilize datasets from both general and specialized domains. For commonsense reasoning, we use LogiQA (Liu et al., 2020), OpenBookQA (Mihaylov et al., 2018), and CosmosQA (Huang et al., 2019). For specialized fields, we construct multi-task benchmarks from professional domains, using datasets such as **medicine** (MedMCQA (Pal et al., 2022)), **law** (AGIEval-Law (Zhong et al., 2023)), and **finance** (FinEval (Zhang et al., 2023)). For each domain, the source task set for Scaffold Discovery was constructed from distinct sub-tasks (e.g., different legal subjects for law, or diverse reasoning types for commonsense), chosen to represent a reasonable diversity of challenges within that domain. All remaining sub-tasks were held out as target tasks for evaluation. This ensures strict separation between the data used for scaffold discovery and for residual learning, and matches the domain construction described in Section 3. We provide per-domain statistics for the number and types of source and target sub-tasks in the appendix.

**Models and Implementation Details**   We employ a wide range of LLMs for optimization and evaluation. The **DiSR** framework itself is powered by PaLM 2 models (Anil et al., 2023), where PaLM 2-L-IT served as the *Proposer LLM* and PaLM 2-L as the *Scorer LLM*. We selected these powerful models for two key reasons: (1) to ensure a fair and rigorous comparison against state-of-the-art baselines, which utilize similarly capable models, and (2) to isolate the effects of our decompositional framework by minimizing the possibility that the optimizer's performance is a confounding variable. While our implementation relies on this specific setup, we hypothesize that the architectural advantages of the DiSR framework are model-agnostic, and leave a systematic study with alternative open-source optimizer models to future work. The optimization process for both stages was run for 100 iterations. The prompts generated by DiSR and the baselines are evaluated on a diverse testbed of external LLMs. These include a mix of leading foundational models, such as GPT-4 (OpenAI, 2023b), the LLaMA series (Touvron et al., 2023; AI@Meta, 2024), and Vicuna (Zheng et al., 2023), alongside numerous domain-specific architectures.

**Baseline Methods**   To contextualize DiSR's performance, we compare it against a manual prompt and the powerful OPRO (Yang et al., 2024) baseline. OPRO was chosen as the primary SOTA baseline due to its competitive standing in the "LLM as optimizer" paradigm, optimizing prompts from a generic starting point. For a fair comparison, OPRO was implemented with the same Proposer and Scorer LLMs as DiSR, using a comparable computational budget (100 iterations), the same initial manual prompt $P_0$, and the same meta-prompt template. We also compare against several complementary APO methods on the medical domain, including Iterative-APE (Zhou et al., 2022), PromptAgent (Wang et al., 2023), and APO (Pryzant et al., 2023); these baselines share the same optimization engine and differ only in their search strategy, allowing us to isolate the contribution of the DiSR decomposition. Detailed results are provided in Appendix B.4.

**Evaluation Metrics**   Our evaluation assesses both task performance and prompt semantics. For task performance, we use standard metrics: Accuracy (ACC), Expected Calibration Error (ECE) (Guo et al., 2017), and Area Under the ROC Curve (AUROC) (Fawcett, 2006). Unless otherwise specified, all performance metrics are evaluated in a 0-shot setting. Confidence scores for ECE and AUROC are derived directly from the model's logits for the chosen answer (Yang et al., 2023b). Lower ECE values indicate better-calibrated confidence estimates, which we use to assess the reliability of different prompt optimization methods. To validate our decomposition hypothesis, we analyze the semantic structure of the generated prompts by encoding them into vectors with all-mpnet-base-v2 (Reimers & Gurevych, 2019) and examining their relationships using cosine similarity and t-SNE visualization (Van der Maaten & Hinton, 2008).

## 4.2   MAIN PERFORMANCE COMPARISON

We establish the effectiveness of **DiSR** by comparing it to a leading monolithic APO baseline, OPRO. As shown in our results on commonsense reasoning (Table 1) and professional domains (Figure 3), DiSR provides consistent improvements in prompt quality, accuracy, and calibration. While the SOTA baseline OPRO is competitive, especially on smaller models, DiSR often outperforms it with lower Expected Calibration Error (ECE) across most settings, indicating more reliable prompts without sacrificing performance. On capable models like GPT-4, DiSR achieves top results across all metrics; for example, on GPT-4 for CosmosQA (Table 1), DiSR reaches a top-tier accuracy of **0.60** with a low ECE of **0.17**.

Further testing on specialized models across three professional domains (Figure 3) shows that DiSR-optimized prompts typically outperform manual and OPRO baselines, with clear gains in the medical, legal, and financial domains. This suggests that DiSR's approach excels at structuring and reusing knowledge in specialized areas—critical for prompt optimization. Additional comparisons with Iterative-APE (Zhou et al., 2022), PromptAgent (Wang et al., 2023), and APO (Pryzant et al., 2023) on MedMCQA (Appendix B.4) show similar improvements, indicating that the gains are not specific to a single APO baseline.

## 4.3   EMPIRICAL EVIDENCE FOR THE SCAFFOLD-AND-RESIDUAL HYPOTHESIS

The central claim of our paper is that effective prompts have a modular structure that can be discovered and exploited. We now present two pieces of evidence in Figure 4 supporting this.

Table 1: Performance comparison on commonsense reasoning benchmarks. We report the mean and standard deviation ($_{\pm\text{std}}$) over 3 random trials. DiSR is evaluated against a manual prompt and the SOTA OPRO baseline. DiSR consistently demonstrates an advantage, particularly in producing well-calibrated prompts (lower ECE), which is a key indicator of reliability. Best results are in **bold**.

| Model | Method | LogiQA | | | OpenBookQA | | | CosmosQA | | |
|---|---|---|---|---|---|---|---|---|---|---|
| | | ACC | ECE | AUROC | ACC | ECE | AUROC | ACC | ECE | AUROC |
| LLaMA-2-7B | Manual | $0.32_{\pm0.009}$ | $0.54_{\pm0.043}$ | $0.38_{\pm0.024}$ | $0.35_{\pm0.028}$ | $0.47_{\pm0.038}$ | $0.43_{\pm0.032}$ | $0.33_{\pm0.019}$ | $0.58_{\pm0.046}$ | $0.42_{\pm0.028}$ |
| | OPRO | $0.31_{\pm0.014}$ | $0.48_{\pm0.031}$ | $0.45_{\pm0.029}$ | $0.37_{\pm0.022}$ | $0.41_{\pm0.034}$ | $0.48_{\pm0.036}$ | $0.35_{\pm0.021}$ | $0.42_{\pm0.027}$ | $0.49_{\pm0.031}$ |
| | **DiSR (Ours)** | $\mathbf{0.35_{\pm0.018}}$ | $\mathbf{0.41_{\pm0.025}}$ | $\mathbf{0.53_{\pm0.026}}$ | $\mathbf{0.38_{\pm0.019}}$ | $\mathbf{0.37_{\pm0.031}}$ | $\mathbf{0.55_{\pm0.033}}$ | $\mathbf{0.39_{\pm0.018}}$ | $\mathbf{0.41_{\pm0.024}}$ | $\mathbf{0.51_{\pm0.029}}$ |
| LLaMA-2-13B | Manual | $0.34_{\pm0.017}$ | $0.47_{\pm0.035}$ | $0.47_{\pm0.021}$ | $0.39_{\pm0.025}$ | $0.46_{\pm0.037}$ | $0.45_{\pm0.028}$ | $0.41_{\pm0.022}$ | $0.46_{\pm0.036}$ | $0.59_{\pm0.024}$ |
| | OPRO | $0.37_{\pm0.021}$ | $0.39_{\pm0.020}$ | $0.50_{\pm0.027}$ | $0.42_{\pm0.026}$ | $0.32_{\pm0.023}$ | $0.53_{\pm0.031}$ | $0.42_{\pm0.023}$ | $0.33_{\pm0.024}$ | $0.57_{\pm0.033}$ |
| | **DiSR (Ours)** | $\mathbf{0.37_{\pm0.016}}$ | $\mathbf{0.31_{\pm0.018}}$ | $\mathbf{0.57_{\pm0.025}}$ | $\mathbf{0.45_{\pm0.022}}$ | $\mathbf{0.27_{\pm0.021}}$ | $\mathbf{0.56_{\pm0.029}}$ | $\mathbf{0.45_{\pm0.019}}$ | $\mathbf{0.30_{\pm0.021}}$ | $\mathbf{0.62_{\pm0.028}}$ |
| LLaMA-3-8B | Manual | $0.38_{\pm0.023}$ | $0.42_{\pm0.028}$ | $0.53_{\pm0.023}$ | $0.43_{\pm0.025}$ | $0.35_{\pm0.026}$ | $0.63_{\pm0.024}$ | $0.46_{\pm0.026}$ | $0.26_{\pm0.025}$ | $0.67_{\pm0.021}$ |
| | OPRO | $0.41_{\pm0.021}$ | $0.36_{\pm0.024}$ | $0.61_{\pm0.022}$ | $0.51_{\pm0.018}$ | $0.26_{\pm0.019}$ | $0.71_{\pm0.023}$ | $0.49_{\pm0.015}$ | $0.20_{\pm0.018}$ | $0.68_{\pm0.024}$ |
| | **DiSR (Ours)** | $\mathbf{0.46_{\pm0.017}}$ | $\mathbf{0.31_{\pm0.022}}$ | $\mathbf{0.67_{\pm0.020}}$ | $\mathbf{0.55_{\pm0.016}}$ | $\mathbf{0.21_{\pm0.015}}$ | $\mathbf{0.75_{\pm0.021}}$ | $\mathbf{0.52_{\pm0.024}}$ | $\mathbf{0.15_{\pm0.012}}$ | $\mathbf{0.71_{\pm0.022}}$ |
| Vicuna-7B | Manual | $0.29_{\pm0.029}$ | $0.64_{\pm0.048}$ | $0.44_{\pm0.032}$ | $0.32_{\pm0.027}$ | $0.49_{\pm0.042}$ | $0.37_{\pm0.034}$ | $0.31_{\pm0.020}$ | $0.51_{\pm0.043}$ | $0.47_{\pm0.029}$ |
| | OPRO | $0.31_{\pm0.025}$ | $0.52_{\pm0.038}$ | $\mathbf{0.45_{\pm0.028}}$ | $0.36_{\pm0.025}$ | $0.45_{\pm0.036}$ | $0.44_{\pm0.031}$ | $0.33_{\pm0.016}$ | $\mathbf{0.46_{\pm0.033}}$ | $0.58_{\pm0.035}$ |
| | **DiSR (Ours)** | $\mathbf{0.34_{\pm0.023}}$ | $\mathbf{0.49_{\pm0.031}}$ | $0.41_{\pm0.034}$ | $\mathbf{0.36_{\pm0.022}}$ | $\mathbf{0.41_{\pm0.032}}$ | $\mathbf{0.51_{\pm0.033}}$ | $\mathbf{0.34_{\pm0.018}}$ | $0.53_{\pm0.039}$ | $\mathbf{0.63_{\pm0.030}}$ |
| Vicuna-13B | Manual | $0.32_{\pm0.028}$ | $0.49_{\pm0.034}$ | $0.45_{\pm0.027}$ | $0.35_{\pm0.029}$ | $0.48_{\pm0.035}$ | $0.49_{\pm0.031}$ | $0.36_{\pm0.021}$ | $0.49_{\pm0.038}$ | $0.55_{\pm0.028}$ |
| | OPRO | $0.35_{\pm0.023}$ | $0.42_{\pm0.028}$ | $0.51_{\pm0.026}$ | $0.40_{\pm0.030}$ | $0.43_{\pm0.029}$ | $0.54_{\pm0.027}$ | $0.40_{\pm0.024}$ | $0.39_{\pm0.025}$ | $0.61_{\pm0.032}$ |
| | **DiSR (Ours)** | $\mathbf{0.40_{\pm0.021}}$ | $\mathbf{0.36_{\pm0.022}}$ | $\mathbf{0.55_{\pm0.023}}$ | $\mathbf{0.44_{\pm0.024}}$ | $\mathbf{0.39_{\pm0.026}}$ | $\mathbf{0.58_{\pm0.025}}$ | $\mathbf{0.42_{\pm0.019}}$ | $\mathbf{0.33_{\pm0.021}}$ | $\mathbf{0.64_{\pm0.026}}$ |
| GPT-3.5-Turbo | Manual | $0.35_{\pm0.019}$ | $0.42_{\pm0.026}$ | $0.61_{\pm0.023}$ | $0.37_{\pm0.022}$ | $0.36_{\pm0.027}$ | $0.58_{\pm0.025}$ | $0.42_{\pm0.025}$ | $0.39_{\pm0.028}$ | $0.61_{\pm0.024}$ |
| | OPRO | $0.38_{\pm0.016}$ | $0.32_{\pm0.021}$ | $0.69_{\pm0.019}$ | $0.46_{\pm0.020}$ | $0.28_{\pm0.022}$ | $0.66_{\pm0.023}$ | $0.47_{\pm0.018}$ | $0.26_{\pm0.021}$ | $0.65_{\pm0.023}$ |
| | **DiSR (Ours)** | $\mathbf{0.42_{\pm0.013}}$ | $\mathbf{0.26_{\pm0.028}}$ | $\mathbf{0.73_{\pm0.017}}$ | $\mathbf{0.50_{\pm0.018}}$ | $\mathbf{0.23_{\pm0.019}}$ | $\mathbf{0.70_{\pm0.020}}$ | $\mathbf{0.51_{\pm0.016}}$ | $\mathbf{0.20_{\pm0.015}}$ | $\mathbf{0.68_{\pm0.020}}$ |
| GPT-4 | Manual | $0.44_{\pm0.015}$ | $0.30_{\pm0.022}$ | $0.69_{\pm0.021}$ | $0.45_{\pm0.019}$ | $0.28_{\pm0.024}$ | $0.75_{\pm0.018}$ | $0.54_{\pm0.021}$ | $0.22_{\pm0.017}$ | $0.64_{\pm0.022}$ |
| | OPRO | $0.48_{\pm0.012}$ | $0.23_{\pm0.016}$ | $0.78_{\pm0.014}$ | $0.54_{\pm0.015}$ | $0.21_{\pm0.017}$ | $0.78_{\pm0.015}$ | $0.56_{\pm0.014}$ | $0.19_{\pm0.013}$ | $0.71_{\pm0.019}$ |
| | **DiSR (Ours)** | $\mathbf{0.53_{\pm0.010}}$ | $\mathbf{0.18_{\pm0.012}}$ | $\mathbf{0.82_{\pm0.011}}$ | $\mathbf{0.59_{\pm0.036}}$ | $\mathbf{0.16_{\pm0.009}}$ | $\mathbf{0.83_{\pm0.013}}$ | $\mathbf{0.60_{\pm0.012}}$ | $\mathbf{0.17_{\pm0.011}}$ | $\mathbf{0.75_{\pm0.016}}$ |

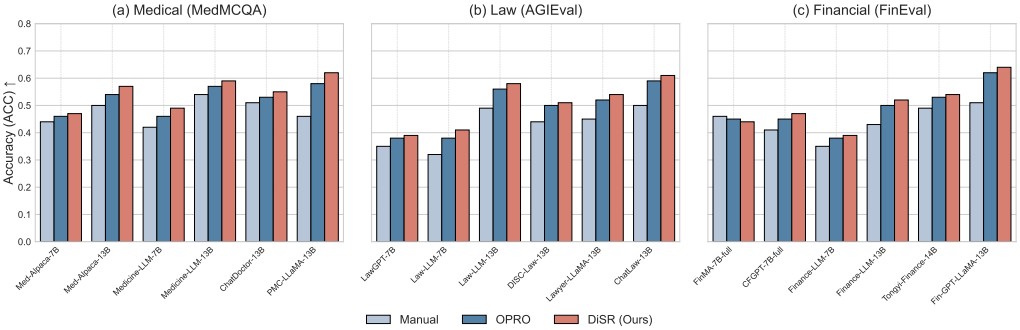

Figure 3: Performance of DiSR against baselines on specialized domain datasets. Each subplot shows the zero-shot accuracy (ACC) for different domains: (a) Medicine (MedMCQA), (b) Law (AGIEval), and (c) Finance (FinEval). DiSR (salmon) often outperforms or matches the SOTA baseline OPRO (blue) and manual prompt (gray), demonstrating the robustness of our decompositional approach in knowledge-intensive domains.

### 4.3.1 SCAFFOLD GENERALIZATION ON UNSEEN TASKS

A key property of a true *Instructional Scaffold* should be its ability to generalize, capturing reusable domain knowledge rather than overfitting to its source tasks. To test this, we perform a leave-one-out validation. For each domain, we run the Scaffold Discovery stage on all but one of its constituent sub-tasks, then apply the resulting $P_{\text{scaffold}}$ in a zero-shot manner to the held-out sub-task.

As shown in Figure 4(a), the discovered scaffolds consistently outperform generic manual prompts. For instance, the medical scaffold boosts the zero-shot accuracy on the unseen MedMCQA task from 0.29 to **0.49** (a +20% absolute improvement), while the finance scaffold provides a +9% gain. This suggests that the Stage 1 process successfully distills not merely a good initialization, but a reusable representation of abstract instructional knowledge for the domain.

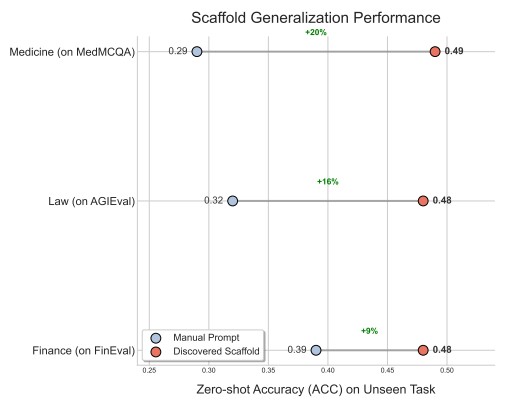

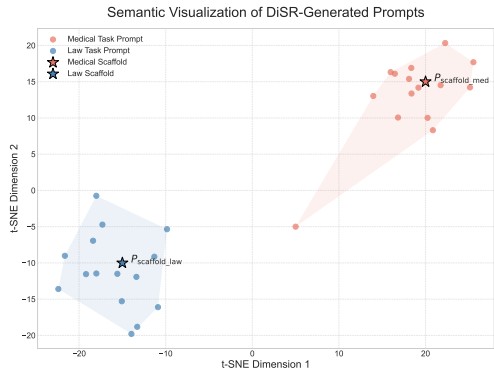

(a) Scaffold Generalization Performance

(b) Semantic Visualization of Prompts

Figure 4: Empirical evidence for the scaffold-and-residual hypothesis. **(a)** The dumbbell plot shows that DiSR's scaffolds consistently outperform manual prompts in zero-shot generalization to unseen tasks. **(b)** t-SNE visualization of prompts reveals clear domain-level clustering, with scaffolds (stars) at the center of each cluster, supporting the modular structure of DiSR-generated prompts.

### 4.3.2 SEMANTIC VISUALIZATION OF THE PROMPT SPACE

To directly visualize the structural hypothesis, we analyze the semantic relationships between prompts generated by **DiSR**. We generate scaffolds and final prompts for two distinct domains (Medicine and Law), encode them into high-dimensional vectors using a sentence transformer, and project them into a 2D space using t-SNE. Figure 4(b) presents the result of this analysis; the visualization reveals a clear structure that is consistent with our theory. We observe:

**Domain-level Clustering:** The prompts form two distinct, well-separated clusters corresponding to the medical and legal domains.

**Scaffold-centered Structure:** At the heart of each domain cluster lies its respective Instructional Scaffold ($P_{\text{scaffold\_med}}$ and $P_{\text{scaffold\_legal}}$). The various task-specific prompts ($P^*$) are distributed around their corresponding scaffold, forming localized neighborhoods in embedding space.

This visualization makes the abstract concepts of scaffolds and residuals more tangible, providing additional evidence that **DiSR** successfully discovers and exploits an underlying modular structure within the prompt space.

### 4.4 ABLATION STUDIES AND FURTHER ANALYSIS

To further probe the properties of our decomposition, we conducted two additional analyses focusing on modularity and efficiency, with results presented in Figure 5.

### 4.4.1 ANALYSIS OF RESIDUAL PORTABILITY

If our decomposition is modular, a residual should be tightly coupled to its corresponding scaffold. We test this via a "transplant" experiment, where we combine a scaffold from one domain with a residual learned for a task in another. As shown in Figure 5(a), the performance of these hybrid prompts drops sharply, scoring lower than correctly composed prompts. These results indicate that a residual is not a generic "performance booster" but a specialized adaptation that is effective when paired with its own scaffold, reinforcing the modularity of the components discovered by DiSR.

### 4.4.2 EFFICIENCY THROUGH AMORTIZATION

The DiSR framework's primary efficiency advantage lies in its amortized optimization model. While our approach requires a one-time, upfront investment for Stage 1 (Scaffold Discovery)—100 iterations in our setup—this cost is rapidly offset by the high efficiency of Stage 2. As illustrated in Figure 5(b), Stage 2 consistently reaches near-optimal performance for each new task in approximately **40 iterations**. This is less than half the computational budget required to run a monolithic

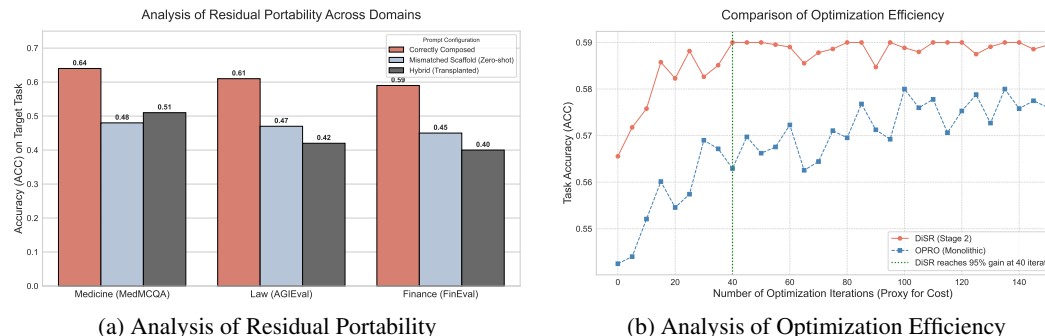

(a) Analysis of Residual Portability      (b) Analysis of Optimization Efficiency

Figure 5: Ablation studies on the emergent properties of the DiSR framework. **(a)** The "transplant" experiment shows performance collapse when a mismatched scaffold and residual are combined (Hybrid), confirming their tight coupling. **(b)** DiSR's Stage 2 (salmon) converges faster and from a higher baseline than monolithic optimization (OPRO, blue), illustrating its efficiency.

method like OPRO from scratch for 100 iterations. Consequently, the initial investment is effectively recuperated after optimizing just the second task, and the average per-task cost becomes significantly lower as more tasks are addressed. This makes DiSR an especially scalable and sustainable approach for prompt optimization in multi-task environments.

### 4.4.3 ANALYSIS OF TASK-SPECIFIC RESIDUALS

To verify the conciseness and semantic role of the residual transformation, we conduct both quantitative and qualitative analyses. Quantitatively, we measure the adaptation's "size" using **token-level edit distance**—the number of tokens added, deleted, or substituted during Stage 2. As shown in Table 2, the resulting adaptations are consistently minimal. The residual's effective length constitutes less than 20% of the scaffold's length across all domains, indicating that DiSR learns a highly *concise* adaptation for each new task.

Table 2: Comparison of component lengths; residuals are consistently more concise than scaffolds.

| Domain | Scaffold Length (tokens) | Avg. Residual Length (tokens) | Residual as % of Scaffold |
|---|---|---|---|
| Medicine | 75 | 7.1 | 9.5% |
| Law | 88 | 15.4 | 17.5% |
| Finance | 52 | 9.8 | 18.8% |

Qualitatively, we conduct a case study to understand the information encoded by the transformation. We examine two related tasks in the medical domain: (A) diagnosing from clinical history, and (B) diagnosing from lab results. While DiSR starts with the same medical scaffold for both, the final prompts differ in key phrases introduced by the transformation. For task (A), it introduces "...based on the following **patient's clinical history**...", while for (B), it adds "...based on the following **laboratory test results**...". This demonstrates that the adaptation is not random noise, but a semantically meaningful process that injects the precise information needed to distinguish between tasks. Taken together, these analyses suggest that DiSR achieves task specialization through minimal, yet semantically precise, modifications to a robust foundational scaffold.

## 5 CONCLUSION

This paper revisits the monolithic paradigm in automatic prompt engineering by introducing a framework based on the Minimum Description Length (MDL) principle. We reconceptualize prompts as decomposable structures, consisting of a reusable **Instructional Scaffold** and a task-specific **Residual**, and propose **DiSR**, a two-stage algorithm to discover these components. Our experiments provide evidence, revealing a scaffold-centered modular structure in the prompt semantic space. This compositional approach enhances optimization efficiency and reliability, transforming prompt engineering from an art of crafting opaque strings to a science of composing structured knowledge.

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

# Appendix

## CONTENTS

## A  EXPERIMENTAL DETAILS

This appendix provides supplementary details regarding the experimental setup, including models, datasets, hyperparameters, and implementation specifics to ensure full reproducibility.

### A.1  EVALUATION MODELS AND DATASETS

**Evaluation Models.** In our experiments, the prompts generated by **DiSR** and all baseline methods are evaluated on a diverse testbed of 25 external large language models. This set is composed of both general-purpose foundational models and models specifically fine-tuned for professional domains, as detailed below.

- **Foundational Models (7):** *GPT-4* (OpenAI, 2023b), *GPT-3.5-Turbo* (OpenAI, 2023a), *LLaMA-3-8B* (AI@Meta, 2024), *LLaMA-2-7B*, *LLaMA-2-13B* (Touvron et al., 2023), *Vicuna-7B*, and *Vicuna-13B* (Zheng et al., 2023).

- **Domain-specific Models (18):** To investigate performance in specialized scenarios, we selected models tailored to three knowledge-intensive domains:
  - *Medicine (6): ChatDoctor-13B* (Li et al., 2023b), *PMC-LLaMA-13B* (Wu et al., 2023), *MedAlpaca-7B*, *MedAlpaca-13B* (Han et al., 2023), *Medicine-LLM-7B*, and *Medicine-LLM-13B* (Cheng et al., 2023).
  - *Law (6): ChatLaw-13B* (Cui et al., 2024), *DISC-LawLLM-13B* (Yue et al., 2023), *Lawyer-LLaMA-13B* (Huang et al., 2023), *LawGPT-7B* (Zhou et al., 2024), *Law-LLM-7B*, and *Law-LLM-13B* (Cheng et al., 2023).
  - *Finance (6): FinGPT-13B-v2* (Yang et al., 2023a), *Tongyi-Finance-14B-Chat*, *FinMA-7B-full* (Xie et al., 2023), *CFGPT-7B-full* (Li et al., 2023a), *Finance-LLM-7B*, and *Finance-LLM-13B* (Cheng et al., 2023).

**Evaluation Datasets.** Our experiments evaluate model performance across six benchmarks. To ensure rigor and reproducibility, we detail the composition of our evaluation sets below.

- *Commonsense Reasoning:* We use the complete official test sets of **LogiQA** (Liu et al., 2020) (651 questions) and **OpenBookQA** (Mihaylov et al., 2018) (500 questions). From the large validation set of **CosmosQA** (Huang et al., 2019), we randomly sampled 1,000 questions.
- *Specialized Domains:* We selected three challenging benchmarks: **MedMCQA** (Pal et al., 2022) (1,000 questions sampled from the test split), **AGIEval** (Zhong et al., 2023) (all questions from the jec-qa-kd and jec-qa-ca law tasks), and **FinEval** (Zhang et al., 2023) (all questions from the validation splits of advanced financial accounting, financial markets, and corporate finance).

## A.2 Hyperparameters and Implementation Details

**Hyperparameters.** The key hyperparameters for the core optimization engine are provided in Table 3. The temperature (0.7) and top-p (0.95) were chosen for a balance of creativity and coherence. Each DiSR stage was run for 100 optimization iterations, which we found empirically sufficient for convergence (see Figure 5(b)). Including 10 historical examples in the Meta-Prompt provided a robust context for iterative refinement.

Table 3: Core hyperparameters for the DiSR optimization loop.

| Hyperparameter | Value |
|---|---|
| *Proposer LLM (PaLM 2-L-IT) Settings* | |
| Temperature | 0.7 |
| Top-p | 0.95 |
| Number of Candidates per Step | 8 |
| *Optimization Loop Settings* | |
| Iterations per Stage (Stages 1 & 2) | 100 |
| Historical Exemplars in Meta-Prompt | 10 |

**Meta-Prompt.** For completeness, we provide the full template and a concrete example of the Meta-Prompt used to guide the Proposer LLM.

---

**Meta-Prompt Template**

—OPTIMIZATION˙HISTORY″″

The subsequent examples illustrate the application of your instruction:
replace ¡INS¿ in each input with your instruction, then read the input
and generate an output. An output is deemed correct if it matches the
provided output; otherwise, it is considered incorrect.

—TASK˙EXAMPLES″″

First, briefly analyze the provided instructions and their scores. What
makes higher-scoring prompts more effective? Based on this analysis,
propose a new, improved instruction that aims to achieve the highest
possible score. Enclose only the new instruction within square brackets.

---

Figure 6: Template for the Meta-Prompt used in the DiSR optimization loop.

## B Additional Results and Analysis

This appendix provides supplementary results and qualitative analyses that further validate and illustrate the findings presented in the main paper.

---

**Example Filled-in Meta-Prompt (Medical Domain)**

Prompt1:
The following are multiple-choice questions (with answers) about
professional medicine. Please choose the correct answer from "A", "B",
"C", or "D".
Score: 61

Prompt2:
Presented below is a multiple-choice question pertaining to professional
medicine. Carefully review the question and select the correct option
from "A", "B", "C", or "D".
Score: 63

The subsequent examples illustrate the application of your instruction:
replace ¡INS¿ in each input with your instruction, then read the input
and generate an output. An output is deemed correct if it matches the
provided output; otherwise, it is considered incorrect.

Input:
A: ¡INS¿
Q: A 13-month-old child is brought to the emergency department because
of urticaria, swelling of the lips, and difficulty breathing
immediately after eating an egg. A potential risk for hypersensitivity
reaction is posed by vaccination against which of the following
illnesses? A: Hepatitis B; Influenza C: Pertussis D: Poliomyelitis
Output: B

First, briefly analyze the provided instructions and their scores. What
makes higher-scoring prompts more effective? Based on this analysis,
propose a new, improved instruction that aims to achieve the highest
possible score. Enclose only the new instruction within square brackets.

---

Figure 7: Instantiated Meta-Prompt example for a medical task during an optimization step.

### B.1 PERFORMANCE IN FEW-SHOT SETTING

To evaluate the robustness of prompts generated by DiSR for in-context learning scenarios, we conducted a five-shot evaluation on the commonsense reasoning datasets. As shown in Table 4, the prompts optimized by DiSR maintain a strong overall performance advantage over both manual and OPRO baselines. While OPRO can occasionally achieve a comparable or superior score on secondary metrics (e.g., ECE for LLaMA-3-8B), DiSR's consistent superiority on the primary accuracy metric highlights that the discovered compositional structures are effective in both zero-shot and few-shot contexts.

### B.2 FULL PERFORMANCE RESULTS ON PROFESSIONAL DOMAINS

This section provides the detailed numerical results for the experiments on specialized professional domains, corresponding to the analysis in Section 4.2 and the visualization in Figure 3. The data, presented in Tables 5, 6, and 7, confirm the trends observed in the main text. Across all professional domains, DiSR consistently delivers the highest accuracy. Furthermore, it shows a strong tendency to produce better-calibrated prompts (lower ECE), although this effect can be part of a trade-off in peak-performance scenarios, as seen with the highly optimized Tongyi-Finance-14B model.

Table 4: Five-shot performance comparison on commonsense reasoning datasets. The DiSR-optimized prompt generally outperforms baselines, demonstrating its robustness in few-shot, in-context learning scenarios. Best performance for each metric is in **bold**.

| Model | Method | LogiQA | | | OpenBookQA | | | CosmosQA | | |
|---|---|---|---|---|---|---|---|---|---|---|
| | | ACC | ECE | AUROC | ACC | ECE | AUROC | ACC | ECE | AUROC |
| LLaMA-2-7B | Manual | $0.37_{\pm0.028}$ | $0.52_{\pm0.042}$ | $0.45_{\pm0.035}$ | $0.40_{\pm0.032}$ | $0.49_{\pm0.041}$ | $0.55_{\pm0.033}$ | $0.37_{\pm0.030}$ | $0.52_{\pm0.044}$ | $0.45_{\pm0.037}$ |
| | OPRO | $0.40_{\pm0.027}$ | $0.44_{\pm0.034}$ | $0.51_{\pm0.031}$ | $0.41_{\pm0.029}$ | $0.46_{\pm0.037}$ | $0.57_{\pm0.031}$ | $0.40_{\pm0.029}$ | $0.49_{\pm0.038}$ | $0.52_{\pm0.032}$ |
| | **DiSR (Ours)** | $\mathbf{0.41_{\pm0.024}}$ | $\mathbf{0.39_{\pm0.030}}$ | $\mathbf{0.55_{\pm0.027}}$ | $\mathbf{0.44_{\pm0.025}}$ | $\mathbf{0.42_{\pm0.034}}$ | $\mathbf{0.61_{\pm0.028}}$ | $\mathbf{0.43_{\pm0.026}}$ | $\mathbf{0.45_{\pm0.036}}$ | $\mathbf{0.56_{\pm0.029}}$ |
| LLaMA-2-13B | Manual | $0.38_{\pm0.025}$ | $0.45_{\pm0.039}$ | $0.49_{\pm0.033}$ | $0.37_{\pm0.032}$ | $0.52_{\pm0.040}$ | $0.41_{\pm0.036}$ | $0.46_{\pm0.028}$ | $0.47_{\pm0.035}$ | $0.63_{\pm0.027}$ |
| | OPRO | $0.41_{\pm0.023}$ | $0.40_{\pm0.030}$ | $0.55_{\pm0.029}$ | $0.47_{\pm0.026}$ | $0.34_{\pm0.031}$ | $\mathbf{0.59_{\pm0.027}}$ | $0.45_{\pm0.025}$ | $0.38_{\pm0.033}$ | $0.62_{\pm0.030}$ |
| | **DiSR (Ours)** | $\mathbf{0.43_{\pm0.022}}$ | $\mathbf{0.37_{\pm0.027}}$ | $\mathbf{0.57_{\pm0.025}}$ | $\mathbf{0.49_{\pm0.023}}$ | $\mathbf{0.31_{\pm0.028}}$ | $0.59_{\pm0.025}$ | $\mathbf{0.48_{\pm0.024}}$ | $\mathbf{0.34_{\pm0.030}}$ | $\mathbf{0.67_{\pm0.026}}$ |
| LLaMA-3-8B | Manual | $0.43_{\pm0.026}$ | $0.35_{\pm0.032}$ | $0.67_{\pm0.023}$ | $0.44_{\pm0.028}$ | $0.30_{\pm0.026}$ | $0.67_{\pm0.024}$ | $0.46_{\pm0.027}$ | $0.25_{\pm0.024}$ | $0.67_{\pm0.025}$ |
| | OPRO | $0.47_{\pm0.021}$ | $0.26_{\pm0.022}$ | $0.77_{\pm0.019}$ | $0.55_{\pm0.021}$ | $0.20_{\pm0.020}$ | $0.81_{\pm0.020}$ | $0.52_{\pm0.022}$ | $\mathbf{0.16_{\pm0.013}}$ | $0.75_{\pm0.020}$ |
| | **DiSR (Ours)** | $\mathbf{0.50_{\pm0.017}}$ | $\mathbf{0.24_{\pm0.021}}$ | $\mathbf{0.79_{\pm0.016}}$ | $\mathbf{0.58_{\pm0.020}}$ | $\mathbf{0.17_{\pm0.018}}$ | $\mathbf{0.84_{\pm0.017}}$ | $\mathbf{0.56_{\pm0.019}}$ | $0.18_{\pm0.019}$ | $\mathbf{0.79_{\pm0.017}}$ |
| GPT-4 | Manual | $0.47_{\pm0.024}$ | $0.26_{\pm0.026}$ | $0.75_{\pm0.020}$ | $0.50_{\pm0.027}$ | $0.21_{\pm0.022}$ | $0.78_{\pm0.021}$ | $0.55_{\pm0.025}$ | $0.18_{\pm0.021}$ | $0.70_{\pm0.023}$ |
| | OPRO | $0.54_{\pm0.018}$ | $0.17_{\pm0.019}$ | $0.85_{\pm0.014}$ | $0.61_{\pm0.017}$ | $0.15_{\pm0.017}$ | $0.84_{\pm0.016}$ | $0.63_{\pm0.017}$ | $0.11_{\pm0.014}$ | $0.86_{\pm0.015}$ |
| | **DiSR (Ours)** | $\mathbf{0.56_{\pm0.015}}$ | $\mathbf{0.14_{\pm0.016}}$ | $\mathbf{0.89_{\pm0.012}}$ | $\mathbf{0.63_{\pm0.016}}$ | $\mathbf{0.12_{\pm0.014}}$ | $\mathbf{0.87_{\pm0.013}}$ | $\mathbf{0.65_{\pm0.015}}$ | $\mathbf{0.10_{\pm0.012}}$ | $\mathbf{0.88_{\pm0.011}}$ |

Table 5: Performance comparison on the **Medicine** domain (MedMCQA dataset).

| Model | Method | ACC ($\uparrow$) | ECE ($\downarrow$) | AUROC ($\uparrow$) |
|---|---|---|---|---|
| Med-Alpaca-7B | Manual | $0.36_{\pm0.032}$ | $0.51_{\pm0.043}$ | $0.47_{\pm0.037}$ |
| | OPRO | $0.38_{\pm0.029}$ | $0.53_{\pm0.041}$ | $0.50_{\pm0.035}$ |
| | **DiSR (Ours)** | $\mathbf{0.43_{\pm0.026}}$ | $\mathbf{0.44_{\pm0.034}}$ | $\mathbf{0.56_{\pm0.030}}$ |
| Medicine-LLM-13B | Manual | $0.52_{\pm0.025}$ | $0.34_{\pm0.028}$ | $0.71_{\pm0.022}$ |
| | OPRO | $0.55_{\pm0.023}$ | $0.30_{\pm0.026}$ | $0.73_{\pm0.019}$ |
| | **DiSR (Ours)** | $\mathbf{0.63_{\pm0.016}}$ | $\mathbf{0.17_{\pm0.020}}$ | $\mathbf{0.74_{\pm0.017}}$ |
| PMC-LLaMA-13B | Manual | $0.47_{\pm0.027}$ | $0.40_{\pm0.033}$ | $0.54_{\pm0.031}$ |
| | OPRO | $0.54_{\pm0.024}$ | $0.25_{\pm0.025}$ | $\mathbf{0.68_{\pm0.014}}$ |
| | **DiSR (Ours)** | $\mathbf{0.56_{\pm0.018}}$ | $\mathbf{0.22_{\pm0.021}}$ | $0.67_{\pm0.019}$ |

Table 6: Performance comparison on the **Law** domain (AGIEval dataset).

| Model | Method | ACC ($\uparrow$) | ECE ($\downarrow$) | AUROC ($\uparrow$) |
|---|---|---|---|---|
| LawGPT-7B | Manual | $0.43_{\pm0.032}$ | $0.45_{\pm0.038}$ | $0.68_{\pm0.026}$ |
| | OPRO | $0.44_{\pm0.029}$ | $0.41_{\pm0.031}$ | $0.67_{\pm0.028}$ |
| | **DiSR (Ours)** | $\mathbf{0.47_{\pm0.024}}$ | $\mathbf{0.39_{\pm0.027}}$ | $\mathbf{0.69_{\pm0.022}}$ |
| Law-LLM-13B | Manual | $0.45_{\pm0.027}$ | $0.24_{\pm0.025}$ | $0.69_{\pm0.021}$ |
| | OPRO | $0.51_{\pm0.023}$ | $0.25_{\pm0.029}$ | $0.71_{\pm0.019}$ |
| | **DiSR (Ours)** | $\mathbf{0.58_{\pm0.018}}$ | $\mathbf{0.18_{\pm0.017}}$ | $\mathbf{0.75_{\pm0.015}}$ |
| ChatLaw-13B | Manual | $0.54_{\pm0.025}$ | $0.38_{\pm0.034}$ | $0.63_{\pm0.030}$ |
| | OPRO | $0.62_{\pm0.018}$ | $0.15_{\pm0.014}$ | $\mathbf{0.73_{\pm0.013}}$ |
| | **DiSR (Ours)** | $\mathbf{0.64_{\pm0.011}}$ | $\mathbf{0.12_{\pm0.010}}$ | $0.72_{\pm0.016}$ |

## B.3 QUALITATIVE EXAMPLES AND ANALYSIS

**Qualitative Examples of DiSR Outputs.** To provide a concrete view of the discovered components, we present qualitative examples for each domain. A consistent pattern emerges: the **Instructional**

Table 7: Performance comparison on the **Finance** domain (FinEval dataset).

| Model | Method | ACC ($\uparrow$) | ECE ($\downarrow$) | AUROC ($\uparrow$) |
|---|---|---|---|---|
| FinMA-7B-full | Manual | $0.44_{\pm 0.034}$ | $0.51_{\pm 0.041}$ | $0.56_{\pm 0.032}$ |
| | OPRO | $0.47_{\pm 0.031}$ | $0.52_{\pm 0.039}$ | $0.61_{\pm 0.030}$ |
| | **DiSR (Ours)** | $\mathbf{0.51_{\pm 0.027}}$ | $\mathbf{0.38_{\pm 0.028}}$ | $\mathbf{0.65_{\pm 0.025}}$ |
| Finance-LLM-13B | Manual | $0.47_{\pm 0.029}$ | $0.31_{\pm 0.027}$ | $0.60_{\pm 0.026}$ |
| | OPRO | $0.53_{\pm 0.023}$ | $0.22_{\pm 0.022}$ | $\mathbf{0.72_{\pm 0.019}}$ |
| | **DiSR (Ours)** | $\mathbf{0.56_{\pm 0.021}}$ | $\mathbf{0.21_{\pm 0.020}}$ | $\mathbf{0.72_{\pm 0.016}}$ |
| Tongyi-Finance-14B | Manual | $0.52_{\pm 0.026}$ | $0.33_{\pm 0.024}$ | $0.66_{\pm 0.023}$ |
| | OPRO | $0.59_{\pm 0.018}$ | $\mathbf{0.20_{\pm 0.014}}$ | $\mathbf{0.76_{\pm 0.012}}$ |
| | **DiSR (Ours)** | $\mathbf{0.61_{\pm 0.010}}$ | $0.22_{\pm 0.017}$ | $0.75_{\pm 0.015}$ |

**Scaffold** establishes the core context, while the learned **Task-specific Residual** (highlighted in bold) injects concise, fine-grained semantics for the specific task.

Table 8: Qualitative example from the **Commonsense Reasoning** domain (LogiQA).

| Component | Generated Prompt Text |
|---|---|
| **Instructional Scaffold** ($P_{\text{scaffold\_commonsense}}$) | Your task is to solve the following reasoning puzzle. Analyze the provided text and question carefully. Select the most logical conclusion from the options "A", "B", "C", or "D". |
| **Final Prompt** ($P_k^*$) for a deductive reasoning task | Your task is to solve the following reasoning puzzle. Analyze the provided text and question carefully. **Identify the key premises and evaluate the validity of the logical chain of inference.** Select the most logical conclusion from the options "A", "B", "C", or "D". |

Table 9: Qualitative example from the **Medical** domain.

| Component | Generated Prompt Text |
|---|---|
| **Instructional Scaffold** ($P_{\text{scaffold\_med}}$) | As an expert medical professional, analyze the following clinical scenario. Your task is to select the single best answer from the options "A", "B", "C", or "D". Your decision should be based on established medical guidelines and evidence-based practice. |
| **Final Prompt** ($P_k^*$) for "Diagnosis from History" | As an expert medical professional, analyze the following clinical scenario. Your task is to select the single best answer from the options "A", "B", "C", or "D". **Focus specifically on the patient's clinical history and symptoms to determine the most likely diagnosis.** Your decision should be based on established medical guidelines and evidence-based practice. |

Table 10: Qualitative example from the **Law** domain.

| Component | Generated Prompt Text |
|---|---|
| **Instructional Scaffold** ($P_{\text{scaffold\_law}}$) | As a legal analyst, you are presented with a case. Your objective is to identify the most accurate legal conclusion from the choices "A", "B", "C", or "D". Your analysis must be grounded in relevant statutes and legal precedents. |
| **Final Prompt** ($P_k^*$) for "Contract Dispute" | As a legal analyst, you are presented with a case. Your objective is to identify the most accurate legal conclusion from the choices "A", "B", "C", or "D". **Evaluate the formation and validity of the contract in question, paying close attention to the elements of offer, acceptance, and consideration.** Your analysis must be grounded in relevant statutes and legal precedents. |

**Analysis of Low-scoring Prompts.** Analyzing ineffective prompts generated during optimization provides insight into the search process. For example, an early-stage prompt like *"Just answer the question."* (Score: 25) is overly simplistic and lacks context. Conversely, a prompt like *"As a legal expert specializing in maritime law, answer..."* (Score: 35) performs poorly by assuming an

Table 11: Qualitative example from the **Finance** domain.

| Component | Generated Prompt Text |
|---|---|
| **Instructional Scaffold** ($P_{\text{scaffold\_fin}}$) | You are a financial analyst. Based on the data provided, select the best financial recommendation from options "A", "B", "C", or "D". Your answer should align with standard principles of financial analysis and risk management. |
| **Final Prompt** ($P_k^*$) for "Equity Valuation" | You are a financial analyst. Based on the data provided, select the best financial recommendation from options "A", "B", "C", or "D". **Your primary task is to perform an equity valuation; consider metrics such as the P/E ratio, discounted cash flow (DCF), and market comparables.** Your answer should align with standard principles of financial analysis and risk management. |

unnecessarily narrow subfield. These examples highlight the importance of discovering a scaffold that is both specific enough to be useful and general enough to cover the domain, reinforcing the core thesis of our work.

### B.4 Additional APO Baselines on MedMCQA

To further isolate the effect of the DiSR decomposition from the underlying optimization engine, we additionally compare DiSR against several complementary APO baselines on the MedMCQA dataset. All methods share the same Proposer/Scorer LLM pair and meta-prompt template as described in Section 4.1, differing only in their search strategy (OPRO, Iterative-APE (Zhou et al., 2022), PromptAgent (Wang et al., 2023), and APO (Pryzant et al., 2023)). Unless otherwise noted, we report average train and test accuracy on MedMCQA using Medicine-LLM-13B (Cheng et al., 2023) as the evaluation model.

Table 12: Average train and test accuracy on MedMCQA for DiSR and additional APO baselines (evaluated with Medicine-LLM-13B).

| Method | Train ACC | Test ACC |
|---|---|---|
| **DiSR (Ours)** | **0.68** | **0.63** |
| OPRO | 0.60 | 0.55 |
| PromptAgent | 0.59 | 0.53 |
| Iterative-APE | 0.57 | 0.52 |
| APO | 0.54 | 0.50 |

The results in Table 12 show that DiSR achieves the highest accuracy on both the optimization tasks (train) and the held-out evaluation set (test). Compared to the strongest non-decompositional baseline OPRO, DiSR improves test accuracy from 0.55 to 0.63 on MedMCQA, while also outperforming Iterative-APE, PromptAgent, and APO. Because all methods share the same optimization engine and differ only in how they structure the search space, these gains provide additional evidence that the scaffold-and-residual decomposition itself is a key driver of DiSR's performance.

### B.5 Impact of Scaffold Diversity

To study how the diversity of source tasks affects scaffold quality, we conduct a controlled experiment across the three professional domains. For each domain, we vary the number of source sub-tasks used in Stage 1 (2, 4, and 6), and evaluate the resulting scaffold in a zero-shot setting on a single held-out target sub-task from the same domain. Table 13 reports the resulting accuracies.

Several trends emerge. First, even with only two source sub-tasks, the discovered scaffold already provides a substantial improvement over a generic manual prompt in all three domains. Second, increasing the number of source sub-tasks from two to four yields a clear gain, indicating that additional diversity helps the scaffold capture more robust, domain-level regularities. Third, the returns begin to diminish beyond four to six sub-tasks: in the finance domain, for instance, the 4-source and 6-source scaffolds achieve comparable performance. This pattern supports our MDL-inspired view that a relatively small but diverse set of source tasks is sufficient to discover a compact, reusable

Table 13: Zero-shot accuracy on a held-out target sub-task as a function of the number of source sub-tasks used during Scaffold Discovery. Results are shown for a representative domain-specific model in each domain.

| Domain | Manual | 2-Source | 4-Source | 6-Source |
|---|---|---|---|---|
| Medicine | 0.29 | 0.42 | 0.47 | **0.49** |
| Law | 0.34 | 0.41 | 0.45 | **0.47** |
| Finance | 0.32 | 0.38 | **0.41** | **0.41** |

Instructional Scaffold, while very large source sets may offer limited additional benefit relative to their optimization cost.

### B.6 NON-MCQA TASK: SENTIMENT CLASSIFICATION ON SST-2

To verify that the scaffold–residual decomposition is not restricted to multiple-choice QA, we conduct a small-scale experiment on a binary sentiment classification task (SST-2). We follow the same DiSR setup as in the main paper: Stage 1 discovers a domain-level scaffold for sentiment analysis from a set of source subsets, and Stage 2 learns task-specific residuals for the final evaluation split. Prompts are evaluated in a zero-shot setting on held-out SST-2 examples. As in our main experiments, we report Accuracy (ACC) and Expected Calibration Error (ECE).

Table 14: Performance on SST-2 sentiment classification (zero-shot). DiSR maintains the pattern observed on MCQA tasks, achieving slightly higher accuracy and lower ECE than both a manual prompt and a monolithic OPRO baseline.

| Model | Method | ACC ($\uparrow$) | ECE ($\downarrow$) |
|---|---|---|---|
| GPT-3.5-Turbo | Manual | 0.80 | 0.32 |
| | OPRO | 0.87 | 0.24 |
| | **DiSR** | **0.91** | **0.16** |
| LLaMA-2-13B | Manual | 0.71 | 0.38 |
| | OPRO | 0.78 | 0.34 |
| | **DiSR** | **0.82** | **0.22** |

**SST-2 Sentiment Classification.** On the SST-2 sentiment classification benchmark, we instantiate DiSR in a non-multiple-choice setting by treating sentiment prediction as a two-way label selection problem (positive vs. negative) and optimizing instructions over source subsets of the training data. As shown in Table 14, DiSR follows the same pattern as on MCQA tasks: for both GPT-3.5-Turbo and LLaMA-2-13B, it achieves higher accuracy than a manual prompt and a monolithic OPRO baseline, while also yielding lower ECE. For GPT-3.5-Turbo, DiSR improves accuracy from 0.80 (manual) and 0.87 (OPRO) to 0.91, and reduces ECE from 0.32 (manual) and 0.24 (OPRO) to 0.16. For LLaMA-2-13B, accuracy increases from 0.71 (manual) and 0.78 (OPRO) to 0.82 with DiSR, accompanied by a drop in ECE from 0.38 and 0.34 to 0.22. These results suggest that the scaffold–residual decomposition is not restricted to multiple-choice QA, but can also improve calibration and performance on short-text classification tasks.

## C LIMITATIONS

We identify several limitations of the DiSR framework, which also suggest promising directions for future work.

**Dependence on Domain Coherence.** The effectiveness of DiSR depends on its central premise: the existence of a coherent, shared structure across source tasks. If the tasks are conceptually unrelated (e.g., medical diagnosis and code generation), Stage 1 may fail to produce a meaningful scaffold. The quality of the discovered scaffold also depends on the *curation* of these source tasks—their

quantity, diversity, and representativeness. While our experiments show success with a small, diverse set of sub-tasks, optimizing strategies for source task selection remains an important direction for future research, and Appendix B.5 only provides an initial exploration of this question. In settings with very few related tasks or strict compute budgets, the one-time cost of Stage 1 may therefore be harder to amortize.

**Interaction with Model Capability.** Our results in Table 1 reveal a nuanced interaction between our decompositional strategy and model scale. DiSR shows a clear advantage on larger models like GPT-4, but its performance can be matched or surpassed by monolithic baselines on smaller models such as Vicuna-7B. We hypothesize that this is because DiSR relies on the model's ability to leverage abstract knowledge encoded in the scaffold. Smaller models may lack the nuanced reasoning capacity needed to fully benefit from this structure, occasionally performing better with a simpler prompt discovered via monolithic optimization. This suggests that the benefits of prompt decomposition are most pronounced when applied to powerful foundation models.

**Scope of Evaluation and Reliability Claims.** Finally, our empirical scope is restricted to standard quantitative metrics such as accuracy and Expected Calibration Error (ECE) on multiple-choice question-answering benchmarks. We do not evaluate other important dimensions of trustworthiness, including robustness, bias, and factual correctness, nor do we cover a broad range of task formats such as open-ended generation. Investigating how scaffold–residual decompositions interact with these broader aspects of reliability and with more diverse task settings is an important direction for future work.

## D Reproducibility Statement

To ensure the reproducibility of our research, all experiments were conducted on publicly available academic benchmarks, with all models and data splits detailed in Appendix A.1. Our primary baseline, OPRO, was implemented under a comparable computational budget for a fair comparison. The DiSR framework is described in detail in Section 3, and for full transparency, a comprehensive list of all key hyperparameters, including the complete Meta-Prompt template, is provided in Appendix A.2. Our evaluation relies on standard metrics (ACC, ECE, AUROC) with a clearly defined protocol, and we believe these details are sufficient for the research community to verify and build upon our findings.

This research adheres to ethical guidelines by exclusively utilizing public, established academic benchmarks for all experiments, ensuring reproducibility and avoiding the use of private or sensitive data. The language models employed were used in a constrained manner, limited to the evaluation of prompts on these specific, non-sensitive question-answering tasks. The primary goal of our work is to enhance not just performance, but also the reliability and efficiency of prompt-based systems. Our findings indicate that the proposed DiSR method yields better-calibrated prompts (lower ECE), contributing to more trustworthy AI, while also improving computational efficiency. While we acknowledge the dual-use nature of foundational AI research, our focus on modularity and interpretability represents a step toward more controllable and aligned systems.

## LLM Usage Statement

In the preparation of this manuscript, Large Language Models (LLMs) were utilized to aid and polish the writing. All authors have reviewed and take full responsibility for the final content and claims presented in this paper, ensuring all information is accurate and correctly cited.

