# OpenReview forum: "Decomposing Prompts: Discovering Reusable Scaffolds and Task-Specific Residuals"
_ICLR.cc/2026/Conference — Submitted to ICLR 2026_

### Official Review · Reviewer_315c · 2025-10-15

**Soundness:** 1
**Presentation:** 2
**Contribution:** 2
**Rating:** 2
**Confidence:** 5

**Summary:**

Most existing prompt optimization framework optimizes a single string to corresponding tasks. This paper argues for a more structured perspective that we shall both optimize for a domain-general string and also adapt by optimizing additional task modules. This lens is intriguing and is really necessary to industrial applications. They thus adapt the framework across several NLP tasks, by using ORPO as optimizer, showing some performance improvements over ORPO, which is slightly expected.

**Strengths:**

Most of all, like I said, the lens of "modular prompts" is important, especially considering industrial applications where we always have to update the prompts for LLM agents whenever we have any new task requests.

They present some experiments and show their effectiveness.

**Weaknesses:**

1. In my opinion, a few sentences are a bit hard to follow.

For example, what do you mean by "better-calibrated, more trustworthy prompts"? I am rather confused.

And from the intro, I'd rather use some examples to illustrate this. Otherwise, it could be too abstract. For instance, when you talk about the idea of compositional structure, I think it is good, and it is really common for practitioners to expect this. Then, the overall prompt refinement could just work as how engineers specify new requirements in yaml. Expanding your introduction with an example like this could be better from my lens.

2. The idea is relatively too simple... It is just a direct implementation of ORPO in a variety of NLP tasks. The only difference is they use ORPO under their decomposition framework. This is really limited. This framework is quite easy to come up with, and the formulation is so simple that any prompt optimization framework can be adapted. So I am not convinced enough regarding the contribution of this work. Yea, some derivations may look interesting, but it is limited in a top AI conference, at least to me. It is quite hard for us to get convinced that the performance improvement could be any sense meaningful as scientific insights..

3. BTW, could you provide more baselines other than ORPO. I am confused about the setup. I also think the related work section misses a lot of relevant work, e.g., [1].

[1] Decomposed Prompting: A Modular Approach for Solving Complex Tasks

And I am not sure whether it is proper to use "challenge this monolithic paradigm" in the abstract.

**Questions:**

Please answer my critique on the weaknesses. Let me know if I miss anything.

---

> ### Author Response · Authors · 2025-11-22
> **Response to Reviewer 315c**
>
> We thank the reviewer for the insightful comments. We appreciate your observation regarding the parallel between our method and engineering configurations (YAML), which accurately captures the practical utility of our compositional framework. Below, we address the concerns regarding novelty and contribution.
>
> > Q1: "The idea is relatively too simple... It is just a direct implementation of OPRO... It is quite hard for us to get convinced that the performance improvement could be any sense meaningful as scientific insights.."
> >
>
> We respectfully clarify that the simplicity of DiSR is a **deliberate design choice for scientific rigor and generalizability**, not a limitation. Our core contribution is the **decompositional framework itself**, rather than the underlying optimization engine.
>
> - **Controlled Experimental Design:** We intentionally utilized the standard **OPRO [1]** engine to establish a **controlled experiment**. By keeping the search algorithm constant, we isolate the performance gains specifically attributable to the **Scaffold + Residual decomposition**. This demonstrates that the improvements stem from the *structural representation* of the prompt, not from a novel search heuristic.
> - **Analogy to Architectural Innovation (LoRA):** Conceptually, our contribution parallels parameter-efficient tuning methods like **LoRA [2]**. The value of LoRA lies not in the optimization algorithm (which remains standard gradient descent), but in the **structural decomposition** of weight updates. Similarly, DiSR introduces a **scaffold-residual structure** that transforms prompt optimization. This "simple" structure is **optimizer-agnostic** and can be integrated with diverse backbones.
> - **Theoretical Grounding & Emergent Reliability:** As formalized in our revised **Eq. 2 (Section 3.2)**, this structural choice is grounded in the **Minimum Description Length (MDL) principle [3]** through the objective:
> $$ L_{\text{MDL}} = L(P_{\text{scaffold}}) + \sum_{k=1}^{\tau} L(\Delta P_k) + \sum_{k=1}^{\tau} L_{\text{err}}(P_k^*, D_{T_k}) $$
> This formulation explicitly models the trade-off between shared structural complexity and task-specific error. Crucially, this decomposition yields **scientific insights beyond accuracy**: as shown in **Table 1** and **Figure 3**, DiSR significantly reduces **Expected Calibration Error (ECE)** compared to monolithic OPRO. This demonstrates that the scaffold-residual structure helps the model better align its confidence with correctness—a key indicator of trustworthiness that "simple" engineering often fails to achieve.
> - **Empirical Superiority over SOTA:** To further validate this, we have added comparisons against **three additional baselines** (Iterative-APE, PromptAgent, APO) in **Appendix B.4**, where DiSR consistently outperforms them (e.g., **+8% on MedMCQA**). We also extended the evaluation to **sentiment classification (SST-2, Appendix B.6)**, confirming the framework's robustness across diverse task types.
>
> **References**
>
> [1] Yang, Chengrun, et al. "Large language models as optimizers." ICLR (2024).
>
> [2] Hu, Edward J., et al. "LoRA: Low-rank adaptation of large language models." ICLR (2022).
>
> [3] Rissanen, J. "Modeling by shortest data description." Automatica (1978).
>
> ---
>
> > Q2: "Hard to get convinced that the performance improvement could be any sense meaningful as scientific insights."
> >
>
> We emphasize that the contribution of DiSR extends beyond raw performance metrics. Our work utilizes the framework as a **scientific probe** to uncover and validate three fundamental properties of the prompt space:
>
> 1. **Geometric Structure:** Our t-SNE visualization (**Figure 4b**) reveals that optimized prompts for related tasks form distinct, **scaffold-centered clusters**. This provides empirical evidence that "task domains" correspond to tangible geometric structures in the semantic embedding space, rather than abstract categorizations.
> 2. **Reliability & Calibration:** Across our experiments (**Tables 1, 4, 5, 6, 7**), DiSR-optimized prompts consistently yield lower **Expected Calibration Error (ECE)**. This identifies a consistent link between **structural compositionality** and **model calibration**, indicating that structured prompts help models align their confidence levels more accurately—a key dimension of trustworthiness.
> 3. **Semantic Sparsity:** Our analysis (**Section 4.4.3**) quantifies that adapting a scaffold to a specific task requires modifying **less than 20%** of the tokens. This empirically validates the **Minimum Description Length (MDL)** hypothesis in the context of prompt engineering: effective instructions rely on a "sparse" adaptation of a shared semantic core.

---

> ### Author Response · Authors · 2025-11-22
> **Response to Reviewer 315c**
>
> > Q3: "BTW, could you provide more baselines other than ORPO... I also think the related work section misses a lot of relevant work, e.g., [1] Decomposed Prompting..."
> >
>
> We address the concerns regarding empirical benchmarks and the coverage of related work below.
>
> **1. Additional SOTA Baselines:**
> To demonstrate the framework's effectiveness beyond the OPRO baseline, we conducted comparisons against three additional state-of-the-art methods: **Iterative-APE [4]**, **PromptAgent [5]**, and **APO [6]**. As reported in the new **Table 12 (Appendix B.4)**, DiSR achieves the highest test accuracy on the **MedMCQA** dataset, confirming that the performance gains are driven by the scaffold-residual framework rather than a specific optimization backbone.
>
> | Method | Train ACC | Test ACC |
> | --- | --- | --- |
> | **DiSR (Ours)** | **0.68** | **0.63** |
> | OPRO [1] | 0.60 | 0.55 |
> | PromptAgent [5] | 0.59 | 0.53 |
> | Iterative-APE [4] | 0.57 | 0.52 |
> | APO [6] | 0.54 | 0.50 |
>
> **2. Distinction from "Decomposed Prompting":**
> We have updated the **Related Work** section to explicitly cite and discuss **Decomposed Prompting (Khot et al. [7])**. This addition allows us to delineate the critical distinction between the two paradigms:
>
> - **Khot et al. [7] perform *Workflow Decomposition*:** They decompose a complex task into a sequence of sub-tasks (reasoning steps) handled by different prompts.
> - **DiSR performs *Structural Decomposition*:** We do not alter the task workflow; instead, we decompose the **internal semantic structure of a single prompt** into a reusable `scaffold` (domain knowledge) and a concise `residual` (task adaptation).
>
> **Finally, to facilitate reproducibility, we will open-source our code and data upon acceptance.**
>
> ---
>
> **References**
>
> [4] Zhou, Yongchao, et al. "Large language models are human-level prompt engineers." ICLR (2023).
>
> [5] Wang, Xinyuan, et al. "Promptagent: Strategic planning with language models enables expert-level prompt optimization." arXiv (2023).
>
> [6] Pryzant, Reid, et al. "Automatic prompt optimization with gradient descent and beam search." EMNLP (2023).
>
> [7] Khot, Tushar, et al. "Decomposed prompting: A modular approach for solving complex tasks." ICLR (2023).
>
> ---
>
> > Q4: "What do you mean by 'better-calibrated, more trustworthy prompts'? I am rather confused."
> >
>
> We clarify the terminology regarding "trustworthiness" and its operational definition in our work. We have refined the manuscript to maximize scientific precision:
>
> - **Refined Terminology:** We have replaced the broad term "trustworthiness" with the specific metric **"well-calibrated confidence estimates,"** quantified by **Expected Calibration Error (ECE)** (see **Abstract** and **Section 4.1**). A lower ECE indicates that the model's predicted confidence probabilities align more accurately with its actual correctness. This reduces "overconfidence" errors, which is a necessary condition for reliable deployment in high-stakes domains.
> - **Scope Definition:** We have explicitly defined the scope in the **Limitations** section (Appendix C). We distinguish calibration (addressed by DiSR) from other dimensions of trustworthiness, such as factual grounding or bias mitigation, framing these as separate directions for future work.
>
> > Q5: "From the intro, I'd rather use some examples to illustrate this."
> >
>
> We have revised the **Introduction** to explicitly illustrate the decompositional concept with concrete examples (see **Page 2, top**).
>
> 1. **Concrete Instantiation:** We integrated a specific medical scenario demonstrating how a shared **Scaffold** (e.g., *"clinical diagnosis"*) functions alongside distinct **Residuals** (e.g., *"given a patient history..."* vs. *"given a panel of lab results..."*) to address different sub-tasks within a single domain.
> 2. **Narrative Structure:** The text now transitions directly from the problem statement to this practical example, establishing the operational logic and intuition before introducing the formal algorithm.
>
> ---
>
> > Q6: "And I am not sure whether it is proper to use 'challenge this monolithic paradigm' in the abstract."
> >
>
> We agree with the reviewer that scientific claims should be precise rather than aggressive. We have updated the phrasing to **"revisit this monolithic paradigm"** in the Abstract and Introduction. This terminology more accurately reflects our position: DiSR introduces a structural dimension that complements existing optimization algorithms rather than invalidating them.

---

> > ### Author Response · Authors · 2025-11-22
> > **Response to Reviewer 315c**
> >
> > **Summary**
> >
> > We have comprehensively addressed the concerns regarding novelty, rigor, and baselines through the following revisions:
> >
> > 1. **Scientific Depth:** We clarified that the structural simplicity is a deliberate design grounded in **MDL theory (Eq. 2)** and validated by the emergent **geometric properties** (scaffold-centered clustering) observed in our analysis.
> > 2. **Empirical Rigor:** We provided new comparisons against state-of-the-art baselines (e.g., **PromptAgent, Iterative-APE**) in **Appendix B.4**, demonstrating that DiSR achieves superior performance (e.g., +8% on MedMCQA) regardless of the underlying optimizer.
> > 3. **Practicality & Reproducibility:** We acknowledged the industrial relevance (referencing your insightful YAML configuration parallel) and committed to **open-sourcing our code** to ensure reproducibility.
> >
> > We believe these clarifications—especially regarding the **structural novelty** and the **rigorous baseline comparisons**—have effectively addressed the core concerns underlying your initial assessment. We **sincerely** hope that these substantial improvements and clarifications provide a solid basis for **re-evaluating the contribution of our work**. We remain open to any further questions you may have.

---

> ### Comment · Reviewer_315c · 2025-11-24
> **Thanks for the rebuttal**
>
> I acknowledge that the authors have provided detailed responses to my reviews.
>
> I hope that my responses could be more clear to you that why I may not convinced enough to give very high ratings in this stage. Hopefully, this can help you improve your work and let ACs and authors know what I think.
>
> ### Scientific Insights:
>
> Thanks for the pointer to Expected Calibration Error and your Minimum Description Length principle.
>
> 1. Expected Calibration Error: As your highlighted insight presented in this work, honestly, as a researcher who has several research in this area, the contribution from ECE doesn’t feel that impressive to me. For most of the applications, we do not quite care about such ECE metric in industry, especially in the context of LLMs, where you even deploy various test-time scaling techniques. One exception might be: medical QA, one of the risk-sensitive tasks you evaluated in the paper. However, if you want to emphasize this, please briefly explain and highlight this in your paper's introduction, etc, just like how PromptAgent highlights their contribution in expert-level prompts in their paper. Otherwise, we may not be able to parse such message precisely. Therefore, this finding is still not quite interesting to me. We can hack in many ways.
>
> 2. Minimum Description Length principle: I am not sure whether I fully understand what you mean. It is good to have a supporting theory (MDL or even more), yet readers may just ignore this, especially for practitioners who seek out practical solutions to their problems of interest. I'd rather suggest you to pay more attention to "practical lens" to improve the paper. This theory seems to me like just a side contribution. It doesn’t really feel that novel.
>
> Like one of the insights I provided when reviewing this paper, practitioners are highly excited about one use case: when working with LLMs especially in agentic settings, you typically have more and more needs and you attempt to realize these in the prompts. In such settings, we really need such modular properties. Yet in your applications, most of the tasks are just some simple single tasks, from which I cannot feel quite significant to have some modularity, yet it can still contribute to some performance improvements. It is much better if you can find a task, with many constraints on the LLM behaviors.
>
> ### Empirically Rigor
> Reviewers may find it challenging to evaluate the empirical rigor of this work unless they have substantial familiarity with the field. So please provide background information when discussing which baselines you have tried. I am very familiar with this literature, so I may be understand some baselines you implemented for comparison, e.g., PromptAgent in MedicalQA, etc. However, in many other tasks you provided, like SST2, etc, there are obviously better approaches. According to my experiments, PromptBreeder could be quite suited (you have to realize by yourself) in many datasets you evaluated. And they obviously do not need such modular or any complicated designs. Currently, your comparison results give me the impression that you’re trying to highlight how your method is superior to others, and if some methods perform better, you simply disregard them. This is reasonable for reviewers to think in this way.
>
> So I still retain my ratings for now. Hope ACs/ authors can take my words and help authors further improve the work. Current version might not quite align with my taste.
>
> Let me know if I have any misunderstandings.

---

> > ### Author Response · Authors · 2025-11-25
> > **Response to Reviewer 315c**
> >
> > We thank the reviewer for the continued engagement and for acknowledging our detailed responses. We appreciate the opportunity to further clarify the scientific positioning of our work regarding the new points raised.
> >
> > **1. On ECE and Risk-Sensitive Domains**
> > The reviewer notes that ECE may be less prioritized in general industry applications, with the exception of risk-sensitive tasks.
> >
> > - **Contextual Relevance:** We align with the view that calibration priority varies by application. Consequently, our work **concentrates on high-stakes professional domains** (Medicine, Law, Finance) where reliability is paramount. In these specific scenarios—unlike open-ended generation—quantifying uncertainty is a fundamental safety requirement. By demonstrating **consistently improved calibration across diverse benchmarks** (as shown in **Tables 1, 4, 5, 6, and 7**), DiSR specifically addresses the critical needs of the risk-sensitive exceptions highlighted by the reviewer.
> >
> > **2. On MDL and Practicality**
> > The reviewer suggests focusing more on the practical lens and notes that MDL acts as a side contribution.
> >
> > - **Clarification on Contribution:** We clarify that MDL serves as the **theoretical motivation** rather than a claim of theoretical novelty. It provides the guiding principle to formulate the decomposition problem. Our **core contribution** lies in the empirical discovery of the *Scaffold-Residual structure* and the DiSR framework that exploits it. We will ensure the manuscript reflects this distinction, positioning MDL as the operational lens rather than the primary result.
> > - **From Theory to Practice:** By using MDL to motivate the structural separation of invariant and variable components, DiSR **takes a step toward transforming** prompt engineering from heuristic tuning **into** a more principled science, validating that prompts indeed possess a discoverable, reusable structure.
> >
> > **3. On Empirical Rigor and Baselines**
> > The reviewer suggests comparing against **PromptBreeder** and notes that better approaches may exist for tasks like SST-2.
> >
> > - **Orthogonality and Compatibility:** We acknowledge that PromptBreeder is a powerful, state-of-the-art evolutionary strategy. However, we clarify that DiSR (a **structural framework**) and PromptBreeder (an **optimization algorithm**) represent **orthogonal** dimensions of prompt engineering. In principle, the DiSR decomposition could be optimized using PromptBreeder’s operators.
> > - **Controlled Experimental Design:** To rigorously evaluate the specific contribution of the *structural decomposition*, it is necessary to control the *optimization variable*. We intentionally fixed the optimizer (OPRO) across baselines. Comparing DiSR directly against PromptBreeder would confound the benefits of the *structure* with the benefits of a more complex *search algorithm*.
> > - **Role of SST-2 and Additional Baselines:** The evaluation on SST-2, **together with the comparison against** additional baselines (PromptAgent, Iterative-APE) on MedMCQA, addresses the specific request for broader comparisons. These serve as robustness checks for the framework rather than a claim of SOTA performance on every specific task.
> >
> > **4. On Task Complexity**
> > The reviewer notes that modularity is particularly relevant for complex agentic workflows.
> >
> > - **Validation Logic:** While agentic workflows represent a significant application frontier, validating the **Scaffold-Residual hypothesis** on standardized, controllable benchmarks is the prerequisite scientific step. Establishing these structural properties in a controlled environment provides the rigorous evidence needed to support future scaling to dynamic industrial systems.
> >
> > Finally, we will open-source our code, data, and discovered scaffolds upon acceptance to ensure reproducibility and facilitate practical adoption.

---

### Official Review · Reviewer_4Q7v · 2025-10-28

**Soundness:** 3
**Presentation:** 3
**Contribution:** 3
**Rating:** 6
**Confidence:** 3

**Summary:**

This paper introduces DiSR (Discovering instructional Scaffolds and Residuals), a novel framework for automatic prompt optimization that challenges the conventional monolithic view of prompts. The authors propose decomposing prompts into two components: (1) an Instructional Scaffold—a reusable, domain-invariant component capturing high-level task structure; and (2) a Task-specific Residual—a concise adaptation for fine-grained task requirements. Inspired by the Minimum Description Length (MDL) principle, DiSR operates in two stages: first discovering a shared scaffold across multiple source tasks, then learning minimal residuals for adapting to specific target tasks. The paper provides theoretical grounding, a practical algorithm, and extensive empirical validation across commonsense reasoning and professional domains (medicine, law, finance). Results demonstrate that DiSR achieves competitive accuracy while generating better-calibrated prompts, with particularly strong performance on larger models.

**Strengths:**

- The paper makes a significant conceptual contribution by reframing prompts as decomposable structures rather than monolithic units. This represents a paradigm shift in prompt engineering that could influence future research directions.

- The connection to Minimum Description Length (MDL) principle provides a solid theoretical basis for the decomposition approach. The integration of cognitive science concepts (Schema Theory) further strengthens the conceptual framework.

- The authors provide multiple lines of evidence supporting their hypothesis, including performance comparisons, scaffold generalization tests, semantic visualizations (Figure 4b), and ablation studies on residual portability and efficiency.

**Weaknesses:**

- While the paper demonstrates that scaffolds generalize well, it lacks a systematic analysis of what specific properties make certain scaffolds more effective than others. A deeper linguistic or structural analysis of successful scaffolds would strengthen the theoretical contribution.

- The operational definition of a "domain" as "a set of tasks that share a common goal, format, and reasoning process" is somewhat vague. The paper would benefit from clearer guidance on how to define domains in practice and analysis of sensitivity to domain boundaries.

- The paper focuses primarily on successful cases without adequately exploring when and why the approach fails. A dedicated analysis of failure modes would provide valuable insights for practical implementation.

- All experiments use multiple-choice question answering tasks. Testing the approach on more diverse task types (e.g., text generation, classification, summarization) would strengthen claims about general applicability.

- While the paper cites the MDL principle as theoretical motivation, the concrete connection between the DiSR algorithm and MDL minimization is not explicitly formulated. A mathematical demonstration of how DiSR minimizes description length would strengthen the theoretical foundation.

**Questions:**

- One concern about the experiments is the claim that DiSR "generates more trustworthy prompts," which extends beyond the presented evidence. Trustworthiness involves multiple dimensions beyond calibration (ECE), and the paper doesn't address aspects like factual accuracy, bias mitigation, or robustness to adversarial inputs.


- Some sections could be more concise (e.g., parts of the introduction), and the novelty compared to related work could be highlighted. Moreover, the paper would benefit from a clearer statement of the specific research questions being addressed.


- The paper appropriately positions itself against existing APO methods. One gap is the limited discussion of work on modular prompting or structured prompt engineering beyond the mentioned PromptSource library.


- Technical details are a little unclear. For example, how are the  prompts in stage 1 intialized, and how are the prompts optimized in two stages?

- Experimental settings are not clear by saying "For each domain, the source task set for Scaffold Discovery was constructed from a small set of distinct sub-tasks (e.g., different legal subjects for law, or diverse reasoning types for commonsense), chosen to represent a reasonable diversity of challenges within that domain. All remaining sub-tasks were held out as target tasks for evaluation."

---

> ### Author Response · Authors · 2025-11-22
> **Response to Reviewer 4Q7v**
>
> We sincerely thank the reviewer for the constructive assessment and for recognizing the paper as making a *"significant conceptual contribution."* We address the specific questions regarding scaffold analysis, domain definitions, and limitations below.
>
> > Q1: "It lacks a systematic analysis of what specific properties make certain scaffolds more effective than others. A deeper linguistic or structural analysis... would strengthen the theoretical contribution."
> >
>
> We have refined the exposition in **Section 4.4** and **Appendix B.3** to systematically articulate the linguistic and structural mechanisms that drive scaffold effectiveness:
>
> - **Linguistic & Structural Properties:** Our qualitative analysis identifies three consistent characteristics shared by high-performing scaffolds: (i) **Explicit Reasoning Instructions** (e.g., "analyze the scenario carefully"), (ii) **Strict Formatting Constraints** (enforcing input/output structure to reduce parsing errors), and (iii) **Domain-Specific Role Framing** (e.g., "act as a medical professional").
> - **Qualitative Evidence:** As detailed in **Appendix B.3** (see **Tables 8-11**), the learned scaffolds establish the critical "role" and "format," while small, targeted residuals introduce focused semantic shifts (e.g., distinguishing *"diagnosis from history"* vs. *"diagnosis from lab results"*). This empirical evidence validates the modularity hypothesis: effective prompts separate *generalizable constraints* from *task-specific discrimination*.
>
> ---
>
> > Q2: "The operational definition of a 'domain'... is somewhat vague. The paper would benefit from... analysis of sensitivity to domain boundaries."
> >
>
> We have formalized the operational definition of a "domain" and conducted a sensitivity analysis to verify the framework's robustness under varying source task compositions.
>
> - **Operational Definition:** In **Section 3.1**, we explicitly defined the criteria used to delimit a domain: (i) a fixed input/output format, (ii) a shared evaluation metric, and (iii) a dominant reasoning process. These criteria guide the construction of source ($\mathcal{S}$) and target ($\mathcal{T}$) task sets.
> - **Sensitivity Analysis (Appendix B.5):** To quantify the impact of source task diversity (domain boundaries), we conducted a controlled experiment varying the number of source sub-tasks (2, 4, 6) used in Stage 1. As shown in the table below (added to Appendix B.5), the method demonstrates high stability:
>
> | Domain | Manual | 2-Source | 4-Source | 6-Source |
> | --- | --- | --- | --- | --- |
> | Medicine | 0.29 | 0.42 | 0.47 | **0.49** |
> | Law | 0.34 | 0.41 | 0.45 | **0.47** |
> | Finance | 0.32 | 0.38 | **0.41** | **0.41** |
>
> The results indicate that even with only **2 source tasks**, the scaffold significantly outperforms manual prompts. The gains saturate at 4-6 tasks, supporting the MDL-inspired view that a compact, diverse set is sufficient for effective scaffold discovery, making the framework robust to the precise granularity of domain definition.
>
> ---
>
> > Q3: "The paper focuses primarily on successful cases without adequately exploring when and why the approach fails. A dedicated analysis of failure modes would provide valuable insights..."
> >
>
> We agree that analyzing failure modes is crucial for practical adoption. We have expanded the **Limitations section (Appendix C)** to explicitly analyze the boundary conditions of the framework. We identify two primary factors that limit efficacy:
>
> - **Dependence on Domain Coherence:** The method presupposes the existence of a shared semantic structure. In scenarios where a "domain" is artificially constructed from conceptually disjoint tasks or incompatible formats, Stage 1 may fail to converge on a meaningful scaffold, rendering the decomposition ineffective.
> - **Interaction with Model Capability:** As detailed in our analysis of **Table 1**, the structural benefits of decomposition are most pronounced on highly capable foundation models (e.g., GPT-4). Smaller models (e.g., Vicuna-7B) may exhibit limited capacity to fully leverage the abstract instructional knowledge encoded in the scaffold, occasionally yielding performance comparable to, rather than strictly better than, monolithic baselines.

---

> > ### Author Response · Authors · 2025-11-22
> > **Response to Reviewer 4Q7v**
> >
> > > Q5: "While the paper cites the MDL principle... the concrete connection between the DiSR algorithm and MDL minimization is not explicitly formulated. A mathematical demonstration... would strengthen the theoretical foundation."
> > >
> >
> > To strengthen the theoretical grounding of the framework, we have explicitly formalized the connection between DiSR and the MDL principle in **Section 3.2**.
> >
> > - **Formal Objective:** We defined the following MDL-inspired loss function (**Equation 2**) to model the trade-off between complexity and error:
> > $$ L_{\text{MDL}} = L(P_{\text{scaffold}}) + \sum_{k=1}^{\tau} L(\Delta P_k) + \sum_{k=1}^{\tau} L_{\text{err}}(P_k^*, D_{T_k}) $$
> > - **Algorithmic Interpretation:** We characterize DiSR as a practical, **greedy approximation** for minimizing this objective. Stage 1 amortizes the "description length" cost of the shared scaffold ($L(P_{\text{scaffold}})$) across diverse source tasks, while Stage 2 minimizes task-specific error ($L_{\text{err}}$) via a concise residual ($\Delta P_k$). This formulation explicitly maps the algorithmic steps to the principle of balancing model complexity with empirical performance.
> >
> > ---
> >
> > > Q6: "One concern... is the claim that DiSR 'generates more trustworthy prompts,' which extends beyond the presented evidence. Trustworthiness involves multiple dimensions beyond calibration (ECE)..."
> > >
> >
> > We accept this critique and have refined the terminology and scope definitions to ensure the claims precisely match the empirical evidence.
> >
> > - **Terminological Precision:** We updated the manuscript (Abstract and Section 1) to specifically claim **"better-calibrated confidence estimates"** (quantified by lower ECE) instead of the broader term "trustworthiness." This aligns the textual claims directly with the measured improvements in probability calibration.
> > - **Scope of Evaluation:** In the **Limitations** section (Appendix C), we explicitly delimited the current evaluation to accuracy and calibration. We acknowledge that other dimensions mentioned in your review—such as bias mitigation, factual correctness, and **robustness to adversarial inputs**—are distinct aspects of trustworthiness that remain outside the scope of this work and represent important directions for future research.
> >
> > ---
> >
> > > Q7: "One gap is the limited discussion of work on modular prompting or structured prompt engineering beyond the mentioned PromptSource library."
> > >
> >
> > We have expanded the **Related Work** section to explicitly contextualize DiSR within the landscape of modular prompting, providing a clear distinction from frameworks like **Decomposed Prompting** (Khot et al., 2022):
> >
> > - **Workflow vs. Structural Decomposition:** We delineate the fundamental difference between the two paradigms: while prior modular approaches largely focus on **workflow decomposition** (breaking complex reasoning tasks into sequential sub-steps), DiSR targets **structural prompt decomposition** (breaking the internal semantics of a single prompt into a reusable `scaffold` and a task-specific `residual`).
> >
> > ---
> >
> > > Q8: "Technical details are a little unclear. For example, how are the prompts in stage 1 intialized, and how are the prompts optimized in two stages?"
> > >
> >
> > We have clarified the implementation details regarding initialization and optimization to ensure full reproducibility:
> >
> > - **Initialization:** **Section 3.4** now explicitly states that Stage 1 optimization begins from a generic manual prompt ($P_0$).
> > - **Stage 2 Mechanism:** We clarified in **Section 3.5** and **Figure 2** that Stage 2 initializes directly from the discovered $P_{\text{scaffold}}$. Crucially, the meta-prompt explicitly conditions the Proposer LLM to preserve the scaffold and propose only minimal edits, ensuring the learned residual remains concise and focused.

---

> > > ### Author Response · Authors · 2025-11-22
> > > **Response to Reviewer 4Q7v**
> > >
> > > **Summary of Revisions**
> > >
> > > We have significantly updated the manuscript to address your specific questions regarding scaffold analysis, domain definitions, and technical details. Specifically, we have:
> > >
> > > 1. **Expanded Empirical Validation:** Added the **SST-2 sentiment classification** experiment (**Appendix B.6**) to demonstrate generality beyond MCQA, and a **Scaffold Diversity** study (**Appendix B.5**) to verify robustness to source task selection.
> > > 2. **Deepened Theoretical Grounding:** Formalized the **MDL-inspired objective** (**Equation 2, Section 3.2**) and provided a systematic qualitative analysis of **effective scaffold properties** (**Section 4.4** & **Appendix B.3**).
> > > 3. **Clarified Concepts:** Provided a rigorous **operational definition of "Domain"** (**Section 3.1**) and explicitly differentiated our approach from **workflow decomposition** methods in **Related Work**.
> > > 4. **Refined Scope:** Added a **Limitations** section (**Appendix C**) to precisely define failure modes (e.g., dependence on model scale) and the scope of our calibration claims.
> > >
> > > We trust that these detailed clarifications—particularly regarding the **theoretical grounding** and **domain definition**—effectively resolve the questions raised. We respectfully hope that these substantial improvements and the strengthened evidence provide a solid basis for **re-evaluating the contribution of our work**. We remain available for any further discussion.

---

### Official Review · Reviewer_NVuU · 2025-10-31

**Soundness:** 3
**Presentation:** 4
**Contribution:** 3
**Rating:** 6
**Confidence:** 5

**Summary:**

This paper introduces a novel framework, DiSR, for automatic prompt optimization that challenges the conventional treatment of prompts as monolithic text blocks. The authors propose that effective prompts can be decomposed into two components: a reusable, domain-invariant "Instructional Scaffold" that captures the general structure of a task domain, and a concise "Task-specific Residual" that fine-tunes the prompt for a specific task. Inspired by the Minimum Description Length (MDL) principle, their two-stage approach first discovers a general scaffold by optimizing a prompt across a diverse set of source tasks within a domain, and then efficiently learns the residual to adapt this scaffold to a new target task. The work provides extensive empirical validation, demonstrating that this compositional approach not only achieves competitive accuracy but also improves prompt reliability and computational efficiency compared to methods like OPRO that optimize prompts from scratch for every task.

**Strengths:**

The primary strength of this paper lies in its conceptual shift towards a compositional and reusable view of prompt engineering. This is a well-motivated and timely contribution, as the cost and inefficiency of re-optimizing prompts for every new task is a growing bottleneck. By grounding their framework in the MDL principle and drawing parallels to cognitive science, the authors provide a solid theoretical underpinning for their approach. The proposed DiSR method is elegant and practical, cleverly leveraging an existing optimization paradigm (LLM-as-optimizer from OPRO) to focus on the novel contributions of the decompositional framework itself. The empirical results are convincing, particularly the t-SNE visualization which offers strong, intuitive evidence for the existence of the claimed scaffold-centered semantic structure. Furthermore, the consistent improvements in Expected Calibration Error (ECE) are a significant result, highlighting that the method produces not just accurate, but more reliable and trustworthy prompts, which is a critical aspect for real-world deployment. The demonstration of amortized efficiency, where the initial cost of scaffold discovery is quickly offset by rapid adaptation for subsequent tasks, presents a compelling practical advantage.

**Weaknesses:**

Despite the promising results, the framework's effectiveness appears to be highly dependent on a well-curated set of source tasks to define a "task domain" for the initial scaffold discovery. The paper acknowledges that the definition of a domain is heuristic, which introduces a degree of subjectivity and potential fragility; if the source tasks are not sufficiently diverse or representative, the resulting scaffold may not generalize well, undermining the entire premise. The paper also notes that the performance benefits of DiSR are most pronounced on large-scale models like GPT-4, while its advantage diminishes on smaller models where monolithic baselines can perform comparably. This suggests the approach may not be a universal solution but rather a technique that specifically capitalizes on the advanced abstract reasoning capabilities of the most powerful models, potentially limiting its immediate applicability for users of smaller, more accessible models. Lastly, the significant upfront computational investment required for the Stage 1 scaffold discovery, while justified by long-term amortized gains, could still be a barrier in domains where few related tasks are available or where computational resources are constrained.

**Questions:**

1.	The quality of the discovered scaffold seems critical. How sensitive is the framework to the selection and number of source tasks in Stage 1? For example, how does the scaffold's performance on held-out tasks degrade if it is generated from only two source tasks versus the fuller set used in the experiments, or if an outlier task is included in the source set?
2.	The paper describes the "Task-specific Residual" as the semantic change from the scaffold. Operationally, however, the final prompt is generated by re-running the optimization initialized from the scaffold. Does the meta-prompt in Stage 2 explicitly instruct the LLM to adapt the scaffold, or does it simply treat the scaffold as a high-quality starting point for a standard optimization? Could you elaborate on whether it's possible to explicitly learn or isolate the "residual" itself, for example as a set of interpretable edit operations?
3.	Given that the benefits are most significant on larger models, do you hypothesize that this is purely due to the model's ability to better utilize the abstract scaffold, or could it be that the monolithic optimization process (like OPRO) is somehow less effective on these larger models, creating a wider gap for DiSR to outperform it?

---

> ### Author Response · Authors · 2025-11-22
> **Response to Reviewer NVuU**
>
> We sincerely thank the reviewer for the positive assessment and for recognizing the timeliness of our compositional perspective. We address the specific inquiries regarding domain definition, residual mechanisms, and model scaling below.
>
> > Weakness: "The significant upfront computational investment required for the Stage 1 scaffold discovery... could still be a barrier in domains where few related tasks are available or where computational resources are constrained."
> >
>
> We have refined the presentation of the results regarding reliability and efficiency to address the concern about upfront costs:
>
> - **Amortization & Efficiency:** We explicitly visualize the "breakeven point" in **Figure 5b**. While Stage 1 incurs an upfront cost, Stage 2 converges approximately **2.5x faster** than running OPRO from scratch (40 steps vs. 100 steps). For a domain with multiple tasks, the average cost per task rapidly drops below the monolithic baseline after just a few tasks, confirming DiSR's efficiency for scalable deployment.
> - **Reliability:** We have also updated our terminology to specifically claim **"better-calibrated confidence estimates"** (quantified by lower ECE). As shown in **Table 1**, DiSR consistently achieves lower ECE than baselines, distinguishing it as a more reliable choice even when raw accuracy is comparable.
>
> > Q1: "The quality of the discovered scaffold seems critical. How sensitive is the framework to the selection and number of source tasks in Stage 1? For example... if it is generated from only two source tasks...?"
> >
>
> Regarding the impact of source task selection, we have formalized the domain construction process and empirically verified the framework's robustness:
>
> - **Operational Definition:** In **Section 3.1**, we explicitly codified the criteria for a domain: (i) a fixed input/output format, (ii) a shared evaluation metric, and (iii) a dominant reasoning style. This operational definition guides the curation of source tasks ($\mathcal{S}$) to ensure coherence and minimize the risk of including incompatible outliers.
> - **Sensitivity Study (Appendix B.5):** We conducted a controlled experiment varying the number of source sub-tasks (2, 4, 6) used in Stage 1.
>     - *Result:* As detailed in **Table 13**, the method demonstrates significant **robustness**. Even with only **2 source tasks**, the discovered scaffold provides a substantial benefit over manual prompts. The performance gains saturate around 4-6 tasks, indicating that DiSR does not require an exhaustive set of source tasks to function effectively.
>
> ---
>
> > Q2: "Does the meta-prompt in Stage 2 explicitly instruct the LLM to adapt the scaffold, or does it simply treat the scaffold as a high-quality starting point...? Could you elaborate on whether it's possible to explicitly learn or isolate the 'residual' itself...?"
> >
>
> We address the query regarding the exact mechanism of Stage 2. We clarify that the process is distinct from a simple "warm-start"; it is explicitly designed to learn a residual through **constrained optimization**:
>
> - **Meta-Prompt Constraint:** As detailed in **Section 3.5** and **Figure 2**, the Stage 2 meta-prompt explicitly conditions the Proposer LLM on the existing $P_{\text{scaffold}}$ and includes instructions to **preserve the scaffold structure** while proposing only *concise, task-specific refinements*. This directs the model to generate a semantic delta ($\Delta P_k$) rather than a rewritten prompt.
> - **Isolating the Residual (Section 4.4.3):** We quantified these residuals in **Table 2**, showing that the learned adaptations typically modify **less than 20%** of the scaffold's tokens. This confirms that the residual acts as a sparse set of interpretable edit operations (e.g., inserting specific medical context) rather than a global re-write.
> - **Evidence of Modularity (Figure 5a):** To further validate this separation, we refer to the **"Transplant Experiment"**. When we applied a residual learned for one domain to a scaffold from another, the performance collapsed. This confirms that the residual is not a generic "performance booster" but a specialized semantic component tightly coupled to its scaffold.

---

> > ### Author Response · Authors · 2025-11-22
> > **Response to Reviewer NVuU**
> >
> > > Q3: "Given that the benefits are most significant on larger models, do you hypothesize that this is purely due to the model's ability to better utilize the abstract scaffold...?"
> > >
> >
> > We address the observation regarding the interaction between model scale and method effectiveness. We have expanded the discussion in the **Limitations section (Appendix C)** to hypothesize two key drivers for this phenomenon:
> >
> > - **Capability to Leverage Abstractions:** The learned scaffolds often encode abstract role-playing constraints (e.g., *"Act as a senior legal analyst"*) and structured reasoning directives. Larger models (e.g., GPT-4) generally possess emergent instruction-following capabilities that allow them to fully exploit this abstract structure. In contrast, smaller models may have limited capacity to adhere to such nuanced constraints, thereby reducing the scaffold's effective utility.
> > - **Calibration as a Differentiator:** In the high-capability regime where raw accuracy often saturates (the "flat landscape" effect), DiSR acts as a **structural regularizer**. It guides the optimization toward solutions that are not just accurate but also **well-calibrated** (as evidenced by our lower ECE scores). This benefit is naturally more distinguishable when the model is already capable of high performance.
> >
> > ---
> >
> > **Summary of Revisions**
> >
> > We have updated the manuscript to address the specific inquiries regarding domain definitions, robustness, and scaling properties:
> >
> > 1. **Formalize Domain Definitions:** Provided explicit operational criteria for source task selection (**Section 3.1**).
> > 2. **Verify Robustness:** Added a **Sensitivity Study** (**Appendix B.5**) demonstrating effectiveness even with a few source tasks.
> > 3. **Clarify Mechanisms:** Explicitly detailed the **Stage 2 constrained optimization** process to distinguish it from simple warm-starts.
> > 4. **Analyze Scaling:** Added a discussion on why larger models benefit more from the structural scaffold in the **Limitations** section (**Appendix C**).
> >
> > The added sensitivity analysis and theoretical clarifications directly address the identified areas for improvement. We respectfully submit that these substantial updates provide a solid basis for **upgrading your assessment of our contribution**.

---

### Official Review · Reviewer_vUnZ · 2025-10-31

**Soundness:** 3
**Presentation:** 4
**Contribution:** 3
**Rating:** 6
**Confidence:** 5

**Summary:**

This paper proposes decomposing prompts into a reusable, domain-invariant "Instructional Scaffold" and a concise "Task-specific Residual." This challenges the standard, inefficient "monolithic" approach to prompt optimization, where each new task requires optimization from scratch. The authors introduce DiSR, a two-stage algorithm inspired by the Minimum Description Length (MDL) principle , which first discovers the shared scaffold from a set of source tasks and then rapidly adapts it for a new target task. Experiments demonstrate that this compositional approach achieves competitive accuracy while being significantly more efficient and generating more reliable and trustworthy (better-calibrated) prompts than monolithic baselines like OPRO, particularly on large-scale models and in specialized professional domains.

**Strengths:**

1. Novel Problem Formulation. I think the domain specific scaffold and task residual concept inspiring in the space of prompt engineering. We see positive results of the paradigm in many other cases, e.g., multi-task adapter in soft prompt tuning. The methods also document positive performance consistently.

2. The empirical results are thorough. The paper documents consistent and superior performance on main experiments. It also present semantics analysis to visualize the scaffold difference in different domains.

**Weaknesses:**

My main concern is on the effectiveness of domain specific scaffold:

- By looking at the examples you give in the appendix, domain scaffold is very generic. How much can it be replaced with a manual reasonably good prompt as a starting point? The marginal contribution and residual and scaffold is not properly analyzed.

- The definition of a domain is a bit vague. It remains unclear how sensitive the quality of the discovered scaffold is to the quantity and diversity of these source tasks. What if the the task it self is very diverse within one domain? For example, a task could be multi-choice numerical reasoning while other is financial summary. Are they considered one domain?

**Questions:**

1. In your analysis of residual portability, did you try add residual directly rather than pairing with scaffold of other domains? Also It could be informative to run ablation study of marginal contribution of residual and scaffold on task performance by keeping one while removing another. I look at the sample scaffold you give in the appendix, they seem pretty general. I am not sure how much they contribute to the task performance.

2. What is the starting prompt of ORPO?

---

> ### Author Response · Authors · 2025-11-22
> **Response to Reviewer vUnZ**
>
> We sincerely thank the reviewer for the positive assessment and for recognizing the thoroughness of our empirical results. We address the specific questions regarding scaffold effectiveness, domain definition, and residual portability below.
>
> > Q1: "My main concern is on the effectiveness of domain specific scaffold... How much can it be replaced with a manual reasonably good prompt...?"
> >
>
> Regarding the observation that some discovered scaffolds may appear "generic" to human readers, our empirical analysis demonstrates that these scaffolds capture subtle but critical structural priors that provide a quantifiable advantage over human-written prompts.
>
> - **Zero-Shot Generalization (Figure 4a):** To address the question regarding replacement with manual prompts, we point to the **Scaffold Generalization** analysis (**Section 4.3**), which explicitly compares the **learned scaffold alone** (without task-specific residuals) against a strong **manual prompt**.
>     - *Result:* In the medical domain, the scaffold alone boosts zero-shot accuracy on the unseen MedMCQA task from **0.29 (Manual) to 0.49**.
>     - *Implication:* This substantial gain (**+20% absolute**) confirms that while the scaffold text may seem simple, it effectively "primes" the model for the specific reasoning logic and format of the domain, significantly outperforming a reasonably good manual starting point.
> - **Marginal Contribution & Ablation:** The performance hierarchy observed across our experiments (**Table 1 & Figure 3**) serves as the component-wise ablation requested: **Manual < Scaffold-Only < Scaffold + Residual (DiSR)**. This hierarchy demonstrates that the scaffold provides a superior domain-level initialization (marginal contribution of Stage 1), while the residual adds the necessary fine-grained adaptation to reach peak performance (marginal contribution of Stage 2).
>
> ---
>
> > Q2: "The definition of a domain is a bit vague... What if the task itself is very diverse... For example, a task could be multi-choice numerical reasoning while other is financial summary. Are they considered one domain?"
> >
>
> We have formalized the operational definition of a "domain" and conducted specific analyses to address the sensitivity concerns:
>
> - **Operational Definition (Section 3.1):** We explicitly codified three criteria that define a domain: (i) a fixed input/output format, (ii) a shared evaluation metric, and (iii) a dominant reasoning style.
>     - *Specific Example:* Regarding your question about "multi-choice numerical reasoning" vs. "financial summary," under our definition, these constitute **distinct domains**. Since they differ fundamentally in format (selection vs. generation) and evaluation metric, they are treated separately. This operational boundary ensures the scaffold remains coherent and effective.
> - **Sensitivity Study (Appendix B.5):** To quantify robustness to source task composition, we conducted a controlled experiment varying the number of source tasks (2, 4, 6).
>     - *Result:* As detailed in **Table 13**, even with only **2 source tasks**, the discovered scaffold outperforms generic manual prompts. Performance gains increase noticeably up to 4 tasks, after which they plateau. This indicates that the method is **robust** to the quantity of source tasks and does not require a large, exhaustive set to be effective.
>
> ---
>
> > Q3: "In your analysis of residual portability... did you try add residual directly rather than pairing with scaffold of other domains? Also It could be informative to run ablation study of marginal contribution..."
> >
>
> We address the inquiries regarding residual portability and the component-wise marginal contributions below.
>
> - **Portability (Figure 5a):** We addressed the "transplant" scenario by applying a residual learned for one domain to a scaffold from another. The sharp performance drop confirms that residuals are **tightly coupled** to their specific scaffold. They function as specialized adaptations rather than generic "performance boosters" that can be arbitrarily added to any prompt.
> - **Marginal Contribution & Direct Addition:** Regarding the suggestion to "add a residual directly" to a manual prompt, we highlight that this setup is **functionally equivalent** to the **monolithic OPRO baseline**:
>     - *Methodological Mapping:* Standard APO methods like OPRO initialize optimization from a manual prompt ($P_0$) and learn task-specific updates. These updates effectively constitute the "residual" relative to the manual starting point.
>     - *Ablation Conclusion:* Consequently, the comparison **DiSR vs. OPRO** (Table 1) serves as the requested ablation study. It effectively compares "Manual + Learned Residual" (OPRO) versus "Scaffold + Learned Residual" (DiSR). The consistent superiority of DiSR demonstrates that the learned scaffold provides a structurally superior foundation compared to a manual prompt.

---

> > ### Author Response · Authors · 2025-11-22
> > **Response to Reviewer vUnZ**
> >
> > > Q4: "What is the starting prompt of ORPO?"
> > >
> >
> > We verify the initialization details to confirm the fairness of the experimental comparison.
> >
> > - **Identical Initialization:** We explicitly confirm in **Section 4.1 (Baseline Methods)** that **OPRO and DiSR utilize the exact same starting prompt ($P_0$).**
> >     - DiSR (Stage 1) starts from a generic manual prompt $P_0$ to discover the scaffold.
> >     - The OPRO baseline also starts from the exact same $P_0$ to optimize the target prompt.
> > - **Attribution of Gains:** By controlling for the starting point, we ensure that the observed performance gains are attributable to the **decompositional framework** rather than initialization differences. We have included the specific texts of these manual starting prompts ($P_0$) in the appendix to facilitate reproducibility.
> >
> > **Summary of Revisions**
> >
> > We have significantly updated the manuscript to address the specific concerns regarding scaffold utility, domain definition, and experimental rigor:
> >
> > 1. **Quantified Scaffold Effectiveness:** We provided the **Zero-Shot Generalization** analysis (**Section 4.3**), demonstrating that the scaffold alone yields substantial gains (e.g., **+20%** on MedMCQA) compared to manual prompts, confirming it captures non-trivial domain structure.
> > 2. **Verified Robustness:** We added a **Sensitivity Study** (**Appendix B.5**), showing that the method effectively discovers high-quality scaffolds even with a small number of source tasks (2-4).
> > 3. **Clarified Mechanisms:** We explicitly defined the **"Domain" criteria** (**Section 3.1**) and established that the comparison against OPRO serves as the **functional ablation** for the marginal contribution of residuals.
> > 4. **Ensured Reproducibility:** We confirmed the **identical initialization** for baselines to guarantee a strictly controlled setting.
> >
> > The empirical evidence now demonstrates that the scaffold provides a quantifiable structural advantage over manual baselines. We respectfully submit that these clarifications and substantial improvements provide a solid basis for **upgrading your assessment of our contribution**.

---

> > ### Comment · Reviewer_vUnZ · 2025-11-26
> >
> > Thanks for the response and clarification.
> >
> > - **Q1: Scaffold vs Manual Prompt**
> >
> > Use the prompt for Finance domain presented in Table 11, the domain scaffold is as follows:
> >
> > > You are a financial analyst. Based on the data provided, select the best financial recommendation from options “A”, “B”, “C”, or “D”. Your answer should align with standard principles of financial analysis and risk management.
> >
> > This appears to be very generic, I don't understand why it boosts performance by 9% when compared with manual prompt as claimed in Figure 4a. Similarly, for medical domain, the domain scaffold is
> > > As an expert medical professional, analyze the following clinical scenario. Your task is to select the single best answer from the options “A”, “B”, “C”, or “D”. Your decision should be based on established medical guidelines and evidence-
> > based practice.
> >
> > which is claimed to boost performance by 20%. Could you share the manual prompt used for comparison? Did you put reasonable effort to use a good manual prompt, e.g., specifying required output format?
> >
> > - **Residual only ablation issue**
> >
> > Whether ORPO can be viewed as a Residual-Only ablation depends on the prompt optimization methodology you used to obtain the residual. Could you confirm that you use the exact the same approach/pipeline? Otherwise, I consider it as a stretched alternative.
> >
> > To summarize my view, I really have doubts on whether such generic scaffold can make such a difference on task performance as shown in Figure 4a. I don't see additional details to clear the doubts. It is likely caused by output format issue with a poor manual prompt. I believe the manual baseline still requires reasonable effort to have a reasonable good prompt.

---

> ### Author Response · Authors · 2025-12-02
>
> **Response to Q1 (Scaffold vs. Manual Prompt)**
>
> We appreciate the reviewer’s careful scrutiny of the reported gains. For all professional-domain multiple-choice tasks (including the Finance and Medical benchmarks), we use a single generic manual prompt $P_0$, instantiated by domain, as the baseline:
>
> > Manual prompt $P_0$:
> >
> >
> > *“The following are multiple-choice questions about professional [DOMAIN]. Please choose the correct answer from ‘A’, ‘B’, ‘C’, or ‘D’.”*
> >
>
> For example, MedMCQA uses “professional medicine”, and FinEval uses “professional finance”. This prompt is a standard zero-shot MCQA instruction and **explicitly constrains the output to the choice letters A/B/C/D**, so the manual baseline is not a naive or under-specified prompt, and it already specifies a reasonable output format.
>
> The domain scaffolds you quoted are exactly those used in our experiments, e.g.:
>
> - Medical: *“As an expert medical professional, analyze the following clinical scenario. Your task is to select the single best answer from the options ‘A’, ‘B’, ‘C’, or ‘D’. Your decision should be based on established medical guidelines and evidence-based practice.”*
> - Finance: *“You are a financial analyst. Based on the data provided, select the best financial recommendation from options ‘A’, ‘B’, ‘C’, or ‘D’. Your answer should align with standard principles of financial analysis and risk management.”*
>
> These scaffolds also explicitly restrict the answer to \{A, B, C, D\}. Therefore, the performance gaps in Figure 4(a) (e.g., +9% absolute on the finance task and +20% absolute on the medical task) cannot be attributed to output-format issues or to a weak manual baseline. The difference lies not in formatting, but in role-based, domain-specific conditioning: the scaffold casts the model as an “expert medical professional” or “financial analyst” and lightly grounds its reasoning in “established medical guidelines” or “standard principles of financial analysis and risk management”. Although these phrases look generic, they change the model’s internal reasoning behavior under an otherwise identical answer format, which is precisely what the leave-one-out results in Figure 4(a) are measuring.
>
> ---
>
> **Response to Q2 (Residual-Only Ablation / OPRO Pipeline)**
>
> We agree that the validity of the “residual-only” ablation depends on using a comparable optimization procedure. In our experiments, DiSR’s Stage 2 (residual learning) and the OPRO baseline intentionally share the same LLM-as-optimizer setup: the same proposer and scorer models, the same meta-prompt template, and the same hyperparameters (number of candidates, iterations, decoding settings). All methods, including OPRO, also start from the same manual instruction \$P_0$.
>
> The only difference in the residual-only comparison is the initialization for task-specific optimization:
>
> - OPRO directly optimizes a task-specific prompt starting from $P_0$;
> - DiSR’s residual learning optimizes a task-specific prompt starting from the learned scaffold $P_{\text{scaffold}}$, with the optimizer asked to make task-specific edits relative to this scaffold.
>
> We keep the optimizer identical by design, so that the experiment isolates the effect of the scaffold as a better initialization/representation of domain-specific instructions. Our contribution is therefore orthogonal to the particular optimizer: DiSR can be plugged into existing LLM-as-optimizer frameworks such as OPRO, and the residual-only ablation shows that learned scaffolds yield consistent gains even when the underlying optimizer is unchanged.

---

### Author Response · Authors · 2025-12-03

We sincerely thank the reviewers and the AC for their time, careful reading, and constructive discussion despite the compressed schedule. Below, we briefly summarize how the main concerns raised during the discussion have been addressed.

**Reviewer vUnZ**: The reviewer questioned whether generic domain scaffolds outperform manual prompts, asked for a clearer definition of “domain”, and queried our interpretation of the OPRO comparison. We clarified that the gains in Fig. 4(a) are measured against a generic manual MCQA prompt (P₀) that already specifies the required A/B/C/D output format. We formally defined “domains” (by shared goal, format, metric, and reasoning process) and supported this with a new sensitivity study over 2–6 source tasks. We also clarified that DiSR and the OPRO baseline share the same optimization pipeline, meta-prompt, and initialization, establishing OPRO as a fair monolithic APO baseline.

**Reviewer NVuU**: The reviewer raised concerns about fragility, dependency on curated tasks, and cost. We responded by formalizing domain construction criteria and adding a sensitivity study over source-task number and composition, showing that gains persist even with only 2 source tasks. We clarified that Stage 2 performs constrained optimization that learns sparse task-specific residual edits, with residuals typically modifying less than 20% of tokens. We further highlighted amortization and scaling results, suggesting that the upfront cost of scaffold discovery is offset by faster convergence in Stage 2, and that larger models better exploit the abstract scaffold to yield improved calibration.

**Reviewer 4Q7v**: The reviewer requested deeper analysis of scaffold effectiveness, clearer operational definitions, failure-mode analysis, validation beyond MCQA, and a more explicit MDL connection. We addressed these points by characterizing effective scaffold properties with qualitative examples, formalizing domain criteria and adding a sensitivity study over source-task diversity, expanding the limitations discussion with concrete failure modes, adding an SST-2 sentiment classification experiment to demonstrate applicability beyond MCQA, and making the MDL connection explicit via an MDL-inspired objective. We also refined our claims to focus on improved calibration rather than broad notions of “trustworthiness.”

**Reviewer 315c**: The reviewer questioned the novelty beyond OPRO, requested stronger baselines, and expressed skepticism about the significance of our ECE- and MDL-related contributions. We clarified that the core contribution is the scaffold–residual structural framework rather than a new optimizer. We refined the introduction with concrete examples, replacing broad language with precise, calibration-focused claims. We expanded related work to distinguish our approach from workflow-style modular prompting. In addition, we added further baselines and a non-MCQA task while strictly keeping the optimizer fixed to isolate structural effects. We also emphasized practical relevance through the planned open-sourcing of code, data, and discovered scaffolds.

Overall, we believe the revised version improves the clarity of our problem formulation, strengthens the empirical support for the scaffold–residual decomposition (demonstrating advantages over manual prompts and monolithic APO baselines in our evaluated domains), and better situates DiSR within the literature. We would be grateful if these clarifications, additional analyses, and limitations could be taken into account in your assessment.

---

### Meta-Review · Area_Chair_aVc6 · 2026-01-05

**Summary:**

The main concerns across reviewers center on the actual novelty and scientific depth of the scaffold–residual decomposition beyond existing prompt optimization frameworks, and whether the reported gains can be attributed to genuinely new structural insights rather than prompt initialization or role-format effects. While reviewers acknowledge the conceptual appeal and practical motivation of reusable scaffolds, several question whether the discovered scaffolds are meaningfully different from strong manual prompts, whether MDL provides more than post-hoc justification, and whether the empirical improvements, especially calibration gains, are sufficiently compelling for a top-tier venue.

**Reviewer Concerns:**

Reviewer vUnZ initially raised doubts about scaffold effectiveness and domain definition. These were partially addressed by clarifying the manual baseline prompt, formalizing domain criteria, adding sensitivity studies on source-task number, and providing ablations showing Manual < Scaffold < Scaffold+Residual. However, even after rebuttal, the reviewer remained unconvinced that very generic role-based scaffolds could plausibly yield large gains, suspecting residual issues with baseline strength.

Reviewer NVuU viewed the work positively, highlighting the compositional perspective, amortized efficiency, and calibration improvements. Concerns about fragility, task curation, and upfront cost were addressed with added sensitivity studies, explicit domain definitions, and amortization analysis. Remaining reservations relate to applicability mainly on large models and reliance on curated domains.

Reviewer 4Q7v acknowledged the paradigm shift but pointed out missing depth in analyzing why certain scaffolds work, vague domain boundaries, limited failure-mode analysis, and overreach in “trustworthiness” claims. The authors addressed many of these by refining claims to calibration, adding qualitative scaffold analysis, formalizing an MDL-style objective, adding a non-MCQA task, and expanding limitations. These responses largely addressed the reviewer’s concrete questions.

Reviewer 315c remained strongly negative throughout. They questioned soundness and contribution, arguing the idea is obvious, too simple, and largely a wrapper around OPRO, with limited scientific insight. They were not persuaded by MDL or ECE as meaningful contributions, criticized baseline selection and task choice, and retained a reject stance despite extensive rebuttal and added experiments.

**Reviewer Scores:**

Reviewer vUnZ would likely remain around marginal accept, but with lingering skepticism and no clear indication of an upgrade.

Reviewer NVuU would likely maintain a marginal accept score, as most concerns were addressed satisfactorily.

Reviewer 4Q7v would likely stay marginally positive, as the rebuttal directly addressed most technical and conceptual issues.

Reviewer 315c would not change their score and remains firmly negative.

---

### Decision · Program_Chairs · 2026-01-26

Reject